# Systems analysis of intracellular pH vulnerabilities for cancer therapy

Erez Persi[1,2], Miquel Duran-Frigola[3], Mehdi Damaghi[4,5], William R. Roush[6], Patrick Aloy[3,7], John L. Cleveland[8], Robert J. Gillies [4] & Eytan Ruppin[9]

A reverse pH gradient is a hallmark of cancer metabolism, manifested by extracellular acidosis and intracellular alkalization. While consequences of extracellular acidosis are known, the roles of intracellular alkalization are incompletely understood. By reconstructing and integrating enzymatic pH-dependent activity profiles into cell-specific genome-scale metabolic models, we develop a computational methodology that explores how intracellular pH (pHi) can modulate metabolism. We show that in silico, alkaline pHi maximizes cancer cell proliferation coupled to increased glycolysis and adaptation to hypoxia (i.e., the Warburg effect), whereas acidic pHi disables these adaptations and compromises tumor cell growth. We then systematically identify metabolic targets (*GAPDH* and *GPI*) with predicted amplified anti-cancer effects at acidic pHi, forming a novel therapeutic strategy. Experimental testing of this strategy in breast cancer cells reveals that it is particularly effective against aggressive phenotypes. Hence, this study suggests essential roles of pHi in cancer metabolism and provides a conceptual and computational framework for exploring pHi roles in other biomedical domains.

[1] School of Physics and Astronomy, Raymond & Beverly Sackler Faculty of Exact Sciences, Tel-Aviv University, 69978 Tel-Aviv, Israel. [2] Center for Bioinformatics and Computational Biology, Institute of Advanced Computer Studies, Department of Computer Science, University of Maryland, College Park, MD 20742, USA. [3] Joint IRB-BSC-CRG Program in Computational Biology, Institute for Research in Biomedicine (IRB Barcelona), The Barcelona Institute of Science and Technology, Barcelona 08028 Catalonia, Spain. [4] Department of Cancer Physiology, Moffitt Cancer Center and Research Institute, Tampa, FL 33612, USA. [5] Department of Oncologic Sciences, Morsani College of Medicine, University of South Florida, Tampa 33612 FL, USA. [6] Department of Chemistry, The Scripps Research Institute, 110 Scripps Way, Jupiter 33458, USA. [7] Institució Catalana de Recerca i Estudis Avançats (ICREA), Barcelona 08010 Catalonia, Spain. [8] Department of Tumor Biology, Moffitt Cancer Center & Research Institute, Tampa, FL 33612, USA. [9] Cancer Data Science Lab, National Cancer Institute, National Institutes of Health, Bethesda, Maryland 20894, USA. These authors contributed equally: Erez Persi, Miquel Duran-Frigola, Mehdi Damaghi. These authors jointly supervised this work: Erez Persi, Eytan Ruppin. Correspondence and requests for materials should be addressed to E.P. (email: erezpersi@gmail.com) or to E.R. (email: eyruppin@gmail.com)

Most cancer cells manifest metabolic adaptations in accord with the Warburg effect[1–3], including increased glucose and nutrient uptake and lactic acid production, even under aerobic conditions, as well as an adaptation to hypoxic and low-nutrient microenvironments[4,5]. Acidification of the extracellular milieu (low pHe) and concomitant intracellular alkalization of the cytoplasm (high pHi) are other hallmarks of cancer, leading to a reverse pH gradient in cancer cells (pHi > 7.2, pHe ~ 6.7–7.1) vs. normal cells (pHi ~ 7.2, pHe ~ 7.4)[6]. This reverse pH gradient relies on increased expression and/or activity of various plasma membrane transporters and acid efflux proteins that control pH homeostasis[7], including monocarboxylate transporters (MCTs), $Na^+$–$H^+$ exchangers (NHEs), and carbonic anhydrases (CAs). Although locally highly diverse, the mean pHe, and oxygen pressure ($pO_2$) both decrease in a highly correlated manner with distance from nearest blood vessels in tumors[8]. This evokes changes in the activity of various transporters promoting intracellular alkalization[9], with an overall significant correlation, yet a non-linear relationship, between the reverse pH gradient and oxygen availability[10].

Notably, the reverse pH gradient is associated with tumor proliferation, invasion, metastasis, aggressiveness, and treatment resistance[5,6,11–14]. Mechanistically, these phenotypes have been ascribed to effects of extracellular acidosis on several processes, including the induction of growth factors (e.g., *VEGF* via *HIF1α*), using secreted lactic acid as a nutrient source[15], suppression of immune surveillance[16–18], and evolutionary selection for acid-resistant malignant cells in the tumor microenvironment[19–21]. Disrupting pH control by inhibiting membrane transporters has been suggested as a therapeutic strategy[22,23], and indeed some membrane transport inhibitors are now in clinical trials[10,24,25]. Moreover, it has been suggested that inhibiting these transporters induces toxic intracellular acidosis[9], and that an alkaline intracellular environment is required for cancer cell survival[26]. However, it is unclear how pHi is coupled to cancer cell growth and metabolism, and if disrupting pHi control could be exploited for therapeutics.

Given the advent of omics-driven personalized metabolic models[27,28] and robust biochemical data of enzyme kinetics, we sought to fill a computational gap and developed a rigorous methodology that infers the pH-dependent activity profiles of metabolic enzymes, and then integrates them into genome-scale metabolic models (GSMMs) of cancer and normal cells. This in silico systems approach allowed us to assess the effects of interfering with pHi on the intracellular metabolic state, and to suggest and experimentally validate a clinically relevant and novel therapeutic strategy to selectively target cancer.

## Results

**Computational pipeline**. Intracellular pH fluctuations affect enzyme activity by modifying protonation states of key catalytic residues and compromising stability of structural folds[29]. Thus, to model the effects of pHi on the metabolic state of cells, it is essential to know the pH-dependent activity profile of each enzyme. Fortunately, elucidating enzymatic pH-activity profiles is a classical task of enzymologists, who need this to optimize the experimental conditions of their assays. This knowledge has been accumulated in the scientific literature over the years and databases like BRENDA[30] are devoted to cataloging it.

To develop a computational pipeline, we first generated pH-activity profiles for metabolic enzymes by extracting from BRENDA the complete record of experimental measurements of the activity of all enzymes at different pH across all taxa. We then defined a pH-activity profile for each enzyme at six critical points corresponding to 0%, 50%, and 100% of maximal enzymatic activity at the acidic and basic sides, respectively (Fig. 1a). To increase coverage of enzymes with missing experimental data, we inferred missing critical pH values based on available data of close homologs, exploiting the fact that pH-activity profiles of enzymes belonging to the same EC category are highly similar between close homologs (Supplementary Figures 1 and 2). This knowledge-based approach was superior to more classical physics-based methods that are focused on predicting pH stability (Supplementary Figure 3). We further predicted any unassigned pH point using linear regression (Supplementary Figure 4), and verified the performance of our predictors using cross validation, as exemplified by the high correlation between the predicted and the experimental pH values across all six critical pH points (Fig. 1b, and Supplementary Figures 5–7). This procedure generated a complete database of pH-activity profiles that can query the profile of any metabolic enzyme using a homology-based search that is readily applicable to any species (Methods and Supplementary Methods). Using this approach, we obtained pH profiles for 76% of the metabolic enzymes in the human proteome. Importantly, the predicted pH optima of enzymes were concordant with the measured pHi of the cellular compartments in which they reside[7,31] (Fig. 1c), where lysosomal and Golgi apparatus enzymes have relatively acidic pH optima (pHi < 7), while mitochondrial and peroxisomal enzymes have relatively alkaline pH optima (pHi > 7.2).

To model the effects of pHi on cell metabolism, we next integrated the inferred pH activity profiles into cell-specific GSMMs of cancer (NCI-60) and normal (HapMap cell line panel) cells, which we recently validated and used to predict anti-migratory and selective cytotoxic cancer targets[27,28]. pH-dependent activity was modeled by modifying the bounds of the permissible flux range of each reaction as a function of the activity of metabolic enzymes catalyzing the reaction at a given pHi according to the inferred pH-activity profiles, such that enzymes with predicted lowered activity have lower bounds (Methods). Using standard constraint-based modeling (CBM) approaches, this allowed us to compute the cellular proliferation rate and uptake/secretion rates of key metabolites as a function of pHi. Cellular organelles were assumed to be well buffered, and thus constraint modeling was only applied to cytosolic metabolic enzymes; nonetheless, the analysis verified this choice as robust (Methods).

**In silico analysis of pH-dependent metabolism**. Applying the pipeline described above, an in-silico analysis of pH-dependent metabolism of cancer and normal cells was performed (Fig. 2). These analyses indicated that at acidic (low) pHi, cancer cell growth rate is reduced vs. that of normal cells, whereas the situation markedly reverses at an alkaline (high) pHi, where growth of cancer cells is augmented (Fig. 2a). Notably, in contrast to normal cells, cancer cell proliferation is predicted to be sustained at alkaline pHi. This behavior is robust to significant perturbations in the reconstructed pH-activity profiles and, importantly, vanishes under random (i.e., wrong) assignment of pH profiles to enzymes (Supplementary Figure 8).

These analyses also predict that the effect of pHi on proliferation is coupled to the metabolic state of cells, whereby lower oxygen consumption and increased glucose uptake rates are observed in cancer cells at high pHi, while at low pHi these adaptations are reversed (Fig. 2b). As oxygen is available to all cells ex vivo, this suggests a fundamental coupling between the Warburg effect and intracellular alkalization in cancer cells, consistent with the understanding that the Warburg effect supports proliferation[2]. In contrast, in normal cells hypoxia and glycolysis are predicted to be independent of pHi, coupled to the

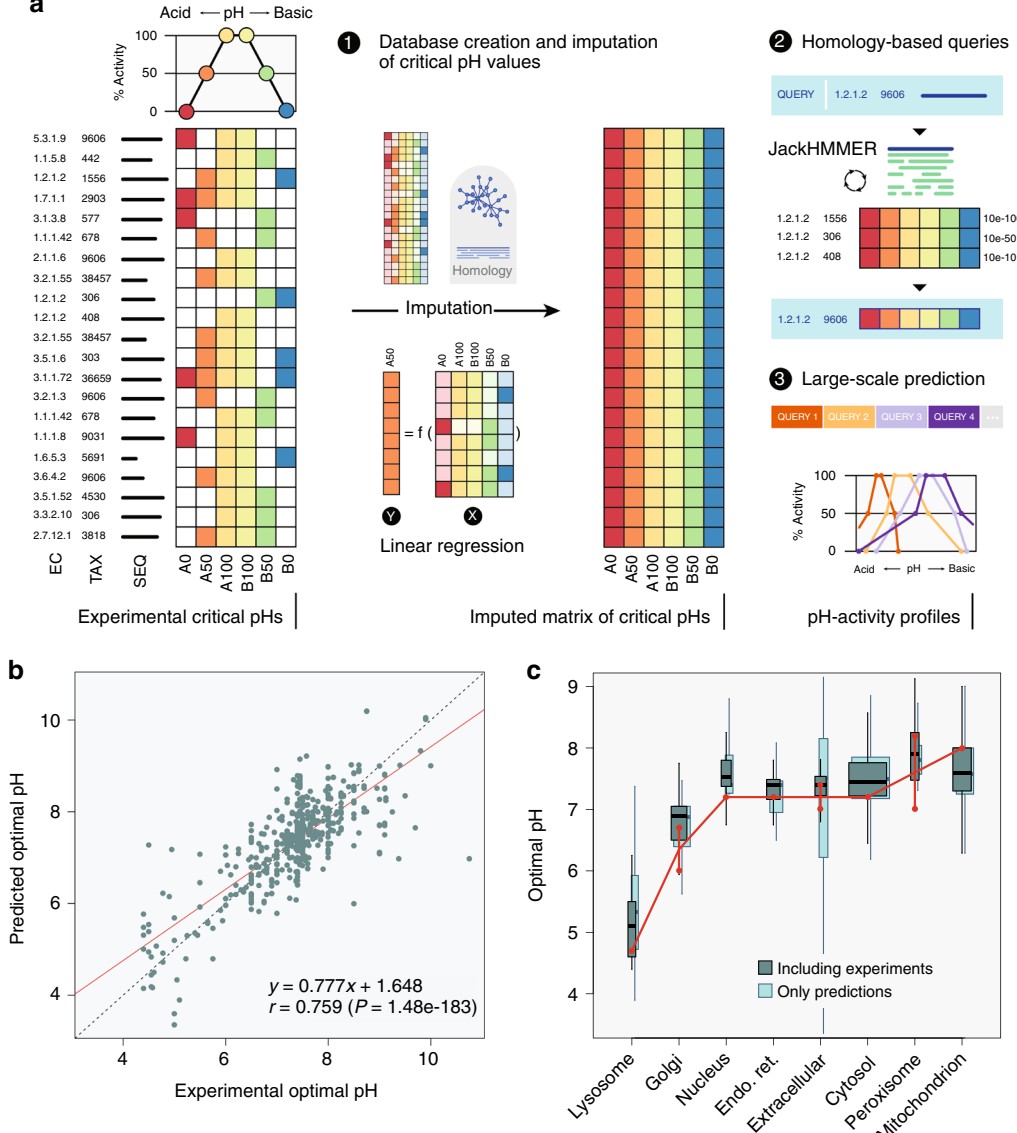

**Fig. 1** Reconstruction of enzymatic pH-dependent activity profiles. **a** Six critical pH points, corresponding to the 0%, 50%, and 100% of enzymatic activity at the acidic and basic regimes were extracted from BRENDA from all taxa. Missing data was complemented with existing data from close homologs or were predicted using linear regressors, generating an imputed database of pH-activity profiles, from which one infers the pH-profile of any enzyme (Methods, Supplementary Methods and Supplementary Figures 1–7 for a complete description). **b** Predicted vs. experimental pH optima, defined as the average of the critical points A100 and B100. The red line depicts linear regression. **c** Distributions of the pH optima of metabolic enzymes in each cellular compartment. Box widths are proportional to the number of enzymes in each compartment. Each box delineates lower quartile, median, and upper quartile values. Most extreme values (whiskers) are within 1.5 times the inter-quartile range from the ends of the box. Red dots depict the measured physiological pHi range of the compartment. "Including experiments" boxes correspond to the pH optima that were used in the subsequent GSMM modeling. As a validation, we include "Only predictions" boxplots, which are the result of the 10-fold cross-validation (Supplementary Figure 5)

weak effect on their proliferation. Finally, in cancer cells these phenotypes strongly correlate with ATP production rate, but not with rates of NADPH production, which is tightly regulated in both cell types. Hence, the sum of these effects predicts that acidifying pHi will selectively impair cancer cell proliferation and reverse the metabolic state of cancer cells to a less fermentative and more oxidative state, presumably with a mild effect on redox. These results are robust to the constraint imposed on proliferation rate (Methods and Supplementary Figure 9).

To identify the most critical targets needed for these metabolic adaptations, a systematic standard divide and conquer search was performed, where pH profiles were applied to increasingly smaller subsets of genes. This analysis identified *GAPDH* (glyceraldehyde

3-phosphate dehydrogenase) and its paralog *GAPDHS*, which catalyze the sixth step (and a principal junction) of glycolysis, as strong modulators of cancer pH-dependent metabolism. Notably, in silico analyses predict that their inhibition selectively augments the effects of intracellular acidosis on cancer cell metabolism and growth (Fig. 2a, b). Thus, the model predicts that inhibition of specific metabolic targets may selectively amplify the anti-proliferative effect of intracellular acidosis on cancer cells, and moreover, that these perturbations may also amplify the anti-Warburg effect of intracellular acidosis on the metabolic state of cancer cells.

To systematically identify putative pH-dependent anti-cancer targets (i.e., anti-proliferative and/or anti-Warburg), the

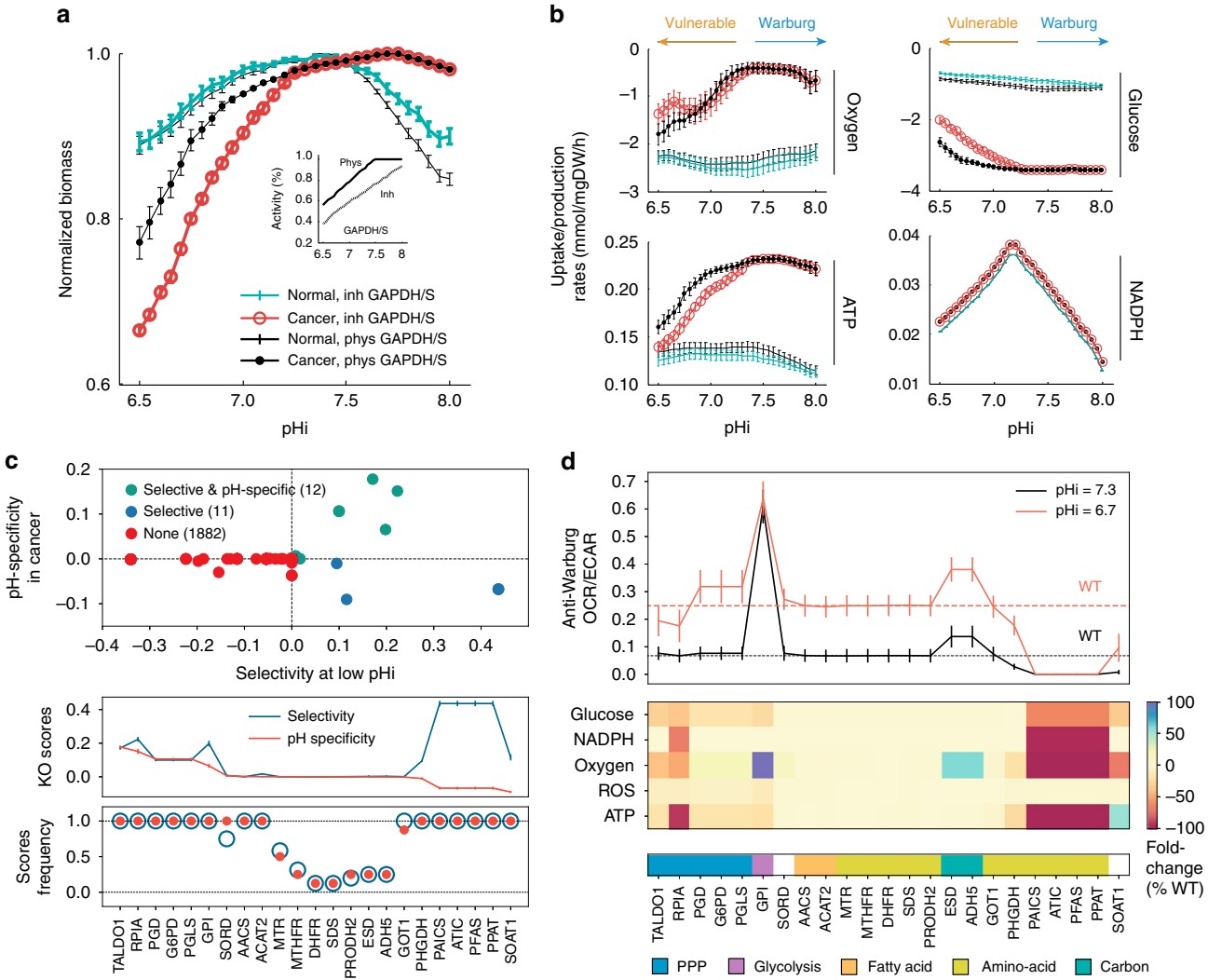

**Fig. 2** In silico pH-dependent metabolism of cancer and normal cell models. **a** Cellular proliferation (biomass yield) as a function of pHi, normalized by the maximal value obtained across all pHi examined, of cancer (circles) and normal healthy (solid) cells, when *GAPDH* is at physiological levels (black) and when it is inhibited (color), as depicted in the inset. **b** Uptake/production rates of oxygen, glucose, total ATP, and total NADPH. Uptake rates are conventionally depicted with a negative sign (more negative values denote higher rates). Error bars depict the standard deviation of the mean values across the populations of GSMMs at each pHi. **c** Anti-proliferative effects of gene inhibition (knockout), showing the classification of knockouts according to their selectivity and pH-specificity scores (top). The predicted targets, ranked by their pH-specificity, with the average selectivity scores superimposed (middle), as well as frequency of scores across all pair comparisons (≥12.5%) are shown (bottom). **d** The anti-Warburg scores (OCR/ECAR) of knockouts at low and physiological pHi (top), and the changes in the uptake/production rates of key metabolites, relative to the wild type (WT) at low pHi (bottom), are shown for each target. Pathways associated with each target are shown in color code. Results are robust with respect to choice of model parameters (Methods and Supplementary Figures 8–12)

metabolic state of cells at physiological (pHi = 7.3) and low (pHi = 6.7) pHi regimes was simulated, and the metabolic consequences of inhibition (knockout (KO)) of each gene (Fig. 2c, d) and each reaction (Supplementary Figure 10) were then assessed. First, we evaluated the anti-proliferative potential of putative targets, across all cancer–normal cell pairs, using two measures (Methods): (i) selectivity, which measures the reduction in proliferation of cancer cells vs. normal cells at low-pHi, where a positive score denotes the selective impaired growth of cancer cells; and (ii) pH-specificity, which measures proliferation rates of cancer cells at low vs. physiological pHi, where a positive score indicates larger inhibition of cancer cells at low pHi. Using these measures 12 enzymes were identified as both selective and pH-specific and 11 targets as selective but not pH-specific (Fig. 2c). The vast majority of enzymes (n = 1882) have no predicted anti-proliferative effect at low pHi (1780 enzymes have zero selectivity

score (SEL), 1839 have zero pH-specificity score (PHS), and the rest have negative scores). The frequency of scores across all examined cancer–normal cell pairs indicates their significance (Methods and Fig. 2c). Interestingly, at the pathway level, we found that targets in the pentose phosphate pathway, glycolysis, and fatty acid metabolism are predicted selective and pH-specific, while most targets involved in amino acid biosynthesis are predicted selective but not pH-specific.

Second, we evaluated the effect of each KO on the production/ consumption rate of key-metabolites, and assessed their anti-Warburg effect by determining the ratio of the oxygen consumption rate (OCR) and the lactate production rate (Fig. 2d). The latter serves as a proxy for the extracellular acidification rate (ECAR). As expected from Fig. 2b, lowering pHi alone reverses the glycolytic and hypoxic nature of wild-type (WT) cancer cells. Relative to WT cancer cells, the additional inhibition of some

identified targets increases the anti-Warburg effect on cancer cells, and this was more significant at low pHi than at physiological pHi (Fig. 2d). Of these, the inhibition of *GPI* has the largest anti-Warburg effect, at both physiological and low pHi. Moreover, the overall predicted mild effect of these KOs on the production rate of reactive oxygen species (ROS) indicates that they are not likely to induce risks associated with excess ROS levels in cancer, namely ROS-induced hypermutation and resistant phenotypes[32]. Interestingly, mitochondrial targets are not prominent in the KO analysis, suggesting the predicted high ATP production rate in cancer (Fig. 2b) is of cytosolic origin.

Complementing the analysis at the gene level with an in silico KO analysis at the reaction level reveals additional targets and confirmed the importance of the reaction catalyzed by *GAPDH*, which is missed by the gene KO analysis due to the existence of the paralog *GAPDHS* (Supplementary Figure 10 and Methods). Lastly, these results were verified to be highly robust at the level of gene inhibition, the exact choice of low pHi, the constraints imposed on proliferation rates (Supplementary Figure 11), and the buffering capacity of cellular compartments (Supplementary Figure 12).

**Experimental proof of concept.** To test the strategy of therapeutically targeting the alkaline pHi preference of cancer cells, experiments were designed to first decrease the pHi by blocking lactate transporters and then inhibit the leading selective and pH-specific targets, *GAPDH* and *GPI* (Fig. 3, Methods, and Supplementary Figures 13–16). This strategy was tested in three breast cell lines, controlling for oxygen availability and pHe levels in the microenvironment: (i) MCF10A normal breast epithelial cells; (ii) naïve and acid-adapted (AA) MCF7 estrogen receptor-positive (ER$^+$) breast cancer cells; and (iii) naïve and AA triple-negative MDA-MB-231 breast cancer cells. These specific AA phenotypes are of particular clinical relevance, as these tumor cells are aggressive in acidic and hypoxic microenvironments[21,33,34] and lack effective therapies. The pHi of cells was determined by confocal microscopy at the single cell level using the emission spectra of the pH-sensitive molecular fluorescent probe SNARF-1 succinimidyl ester (Supplementary Figure 13).

To manipulate pHi we tested the effect of a selective small molecule inhibitor of the MCTs 1 and 2 (MCT1/2)[35] in four different conditions that represent extracellular states of the tumor microenvironment, physiological pHe, low pHe, normoxia, and hypoxia (Fig. 3a). Under normoxia and physiological pHe, the pHi of MCF7 breast cancer cells is at physiological levels and inhibition of MCT1/2 only slightly reduced pHi. Acute hypoxia however significantly reduced pHi, where switching to glycolysis as the only source of energy in the absence of oxygen produces abundant protons as a byproduct, presumably imposing significant stress on the cells. In contrast, chronic hypoxia did not lead to acidic pHi levels; presumably imposing less stress on the cells due to some adaptations, and under these conditions inhibition of MCT1/2 significantly reduced the pHi. In AA MCF7 cancer cells chronic hypoxia plus MCT1/2 inhibition achieved a reduction of pHi by over 0.3 pH units. These effects were weakly dependent of pHe levels (Supplementary Figure 13).

To validate the anti-proliferative effects of the predicted selective and pH-specific targets *GAPDH* and *GPI*, we knocked down each gene using siRNAs (Fig. 3b) and their effects on both cell proliferation (Fig. 3c) and survival (Fig. 3d) were assessed in the four microenvironmental conditions. Consistent with model predictions, reducing pHi via hypoxia and MCT1/2 inhibition impaired the proliferation of both naïve and AA MCF7 cells, and knockdown of *GAPDH* and *GPI* further reduced proliferation in cells, and this was most profound at acidic pHi (Fig. 3c). The

inhibition of *GAPDH* elicited a larger detrimental effect than that of *GPI*, and this was associated with the efficiency of knockdown (Fig. 3b).

To test if the effects on cell proliferation translate into effective killing of cancer cells, we assessed the survival of cells using viability assays (Methods). Notably, reducing pHi via hypoxia plus MCT1/2 inhibition compromised the survival of MCF7 breast cancer cells (Fig. 3d). Under normoxia, where pHi levels remained at physiological levels, cell survival was weakly affected by MCT1/2 inhibition, except for AA cells at low pHe. Further, under acidic pHi conditions, provoked by hypoxia and MCT1/2 inhibition, the additional knockdown of *GAPDH*, and to a lesser extent of *GPI*, triggered cell death, especially in more aggressive AA MCF7 breast cancer cells (Fig. 3d). Extracellular acidosis weakly affected the survival of cancer cells, as expected from the measured weak coupling between pHe and pHi (Supplementary Figure 13). Further, consistent with the model, the strategy is selective for cancer cells, as there were only very modest effects of hypoxia, MCT1/2 inhibition, and knockdown of *GAPDH* or *GPI* on the survival of normal MCF10A breast epithelial cells (Supplementary Figure 14). However, the strategy fails to kill triple negative MDA-MB-231 breast cancer cells, where a sufficiently low pHi was unattainable (Supplementary Figure 15), presumably due to the elevated expression of the *MCT4* transporter that is resistant to the effects of the selective MCT1/2 inhibitor[35,36].

To understand the differences between naïve and AA MCF7 breast cancer cells, we measured the expression of different lactate transporters (Supplementary Figure 16). While naïve cells expressed *MCT2*, only AA cells expressed *MCT4*, and to a lesser extent *MCT1*, across all conditions. Moreover, the metabolic state of AA cells is less fermentative and more oxidative than naïve cells, as exemplified by their lower ECAR and higher OCR rates (Supplementary Figure 16). Hence the successful application of the strategy to these AA cells is likely due to the lowest pHi obtained (~6.9), and possibly also due to their adaptation and reliance on alkaline pHi that renders them more vulnerable to these perturbations, despite the activity of additional transporters and their shift towards oxidative metabolism.

As hypoxia may impose stress on cells and elicit network-wide effects that are hard to control and measure, additional experiments were performed to assess the robustness of the results in naïve and AA MCF7 cells (Fig. 4). In these experiments pHi under normoxia was reduced by inhibiting the function of the Na$^+$–H$^+$ exchanger NHE1 via treatment with cariporide. This strategy was most successful at low pHe (Fig. 4a). We then inhibited the top targets (*GAPDH* and *GPI*), as well as three additional targets identified from different metabolic pathways (*RPIA*, *ACAT2*, and *PFAS*). The efficiency of siRNA-directed knockdown of these targets was verified by qRT-PCR and western blot analyses (Fig. 4b). Also in these experiments, knockdown of these targets compromised the survival of MCF7 cancer cells at low pHi, and again this was more significant at low pHe, where the lowest pHi was achieved (Fig. 4c). Across the different conditions and cells, the predicted selective and pH-specific targets *GAPDH*, *GPI*, and *ACAT2* achieved the largest detrimental effect on cancer cell survival. Relative to control, this was more pronounced in AA cells, despite the smaller reduction in pHi in these cells, suggesting these targets play important roles in the aggressive phenotypes manifest in these cells. In contrast, the predicted selective and pH-specific target *RPIA* displayed inconsistent effects across cells, possibly because it requires larger reduction in pHi to induce anti-proliferative effects in AA cells. As an important control, knockdown of the predicted selective but not pH-specific target *PFAS* did not amplify the anti-proliferative effect relative to control cells at low pHi, and *PFAS* is

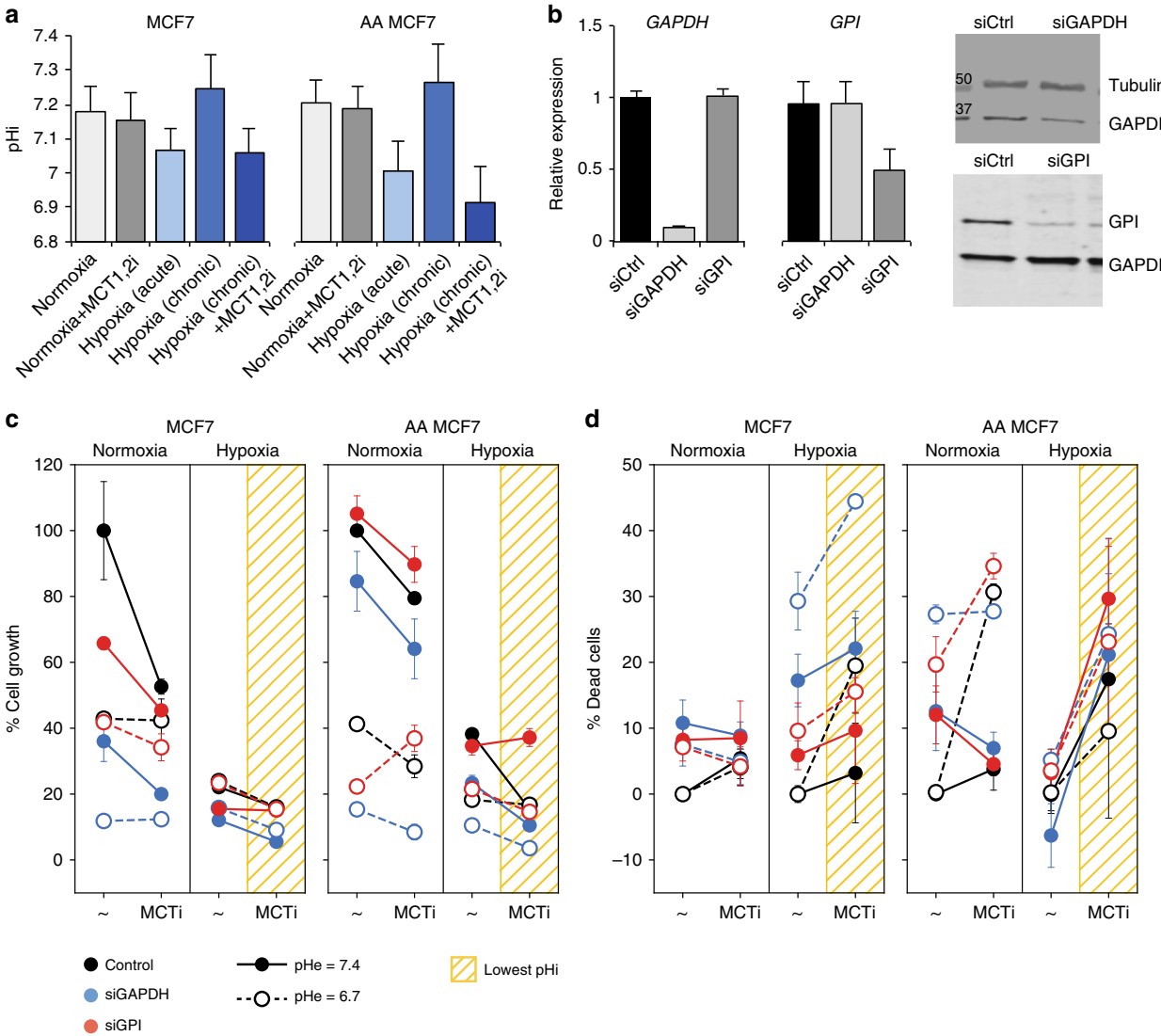

**Fig. 3** Experimental proof of concept. **a** pHi measurements of naïve and acid-adapted (AA) MCF7 breast cancer cells, under different oxygen availability, and following inhibition of *MCT1/2*. For pHi measurments at least 30 cells were analyzed. **b** Efficiency of knockdown of *GAPDH* and *GPI* mRNA and protein following transfection of MCF7 cells with the respective siRNAs (72 h). PCR was done in three separate biological replicate with three repeats. **c** Proliferation assays of cells (Methods) transfected with siRNA targeting *GAPDH* and *GPI*, across four conditions: normoxia, hypoxia, each at physiological pHe (7.4), or acidic pHe (6.7). Drop in proliferation following inhibition of *MCT1/2* is shown as connected lines. Lowest values are obtained when pHi is low (yellow grids). The amplified effect of gene inhibition is seen relative to control (color vs. black). Viability assays is done in three replicates and three reads for each time point. **d** Viability assays (Methods) demonstrate that when pHi is sufficiently low the predicted strategy is particularly effective against AA cancer cells. The inhibition of *GAPDH* and *GPI* achieve efficient killing of cells at low pHi (yellow grids). Note the large slopes obtained for AA cells. The strategy is selective (Supplementary Figure 14), however, fails when sufficiently low pHi is unattainable, as in the case of triple negative breast cancer cells (Supplementary Figure 15). Bars depict the error of the mean over replicas. The experiments were repeated three times with three replicas for each condition

the least pHi-sensitive metabolic target. Finally, we also tested the effects of metabolic perturbations on normal MCF10A breast epithelial cells, where NHE1 inhibition only mildly affected pHi and the viability of cells (Supplementary Figure 17).

To validate the anti-Warburg effect of lowering pHi and of inhibiting selected pHi-dependent targets, we performed Seahorse XF assays and measured the anti-Warburg ratio (OCR/ECAR) in MCF7 breast cancer cells (Fig. 5). These measurements were performed in normal pHe, because of technical difficulties of Seahorse assays to perform well in low pHe. Lowering pHi alone was revealed to have an anti-Warburg effect on cancer cells, consistent with our computational results (cf., Fig. 2b). Further, the knockdown of each of the pHi-dependent targets amplified

the anti-Warburg effect on cancer cells at physiological pHi, and this amplification was more significant at low pHi, following NHE1 inhibition, also consistent with our model (cf., Fig. 2d). These increased anti-Warburg effects at low pHi were not observed in normal MCF10A cells, which overall have higher OCR/ECAR ratio (Supplementary Figure 17).

Nonetheless, while the model predicted an amplified effect relative to control cells for only a few targets, the experiments show that all of the targets examined exhibited an amplified anti-Warburg effect, across all conditions. Thus, intracellular acidosis has a stronger anti-Warburg effect than that predicted by the model. This discrepancy may be due to a number of differences between the model and experiments, as discussed below.

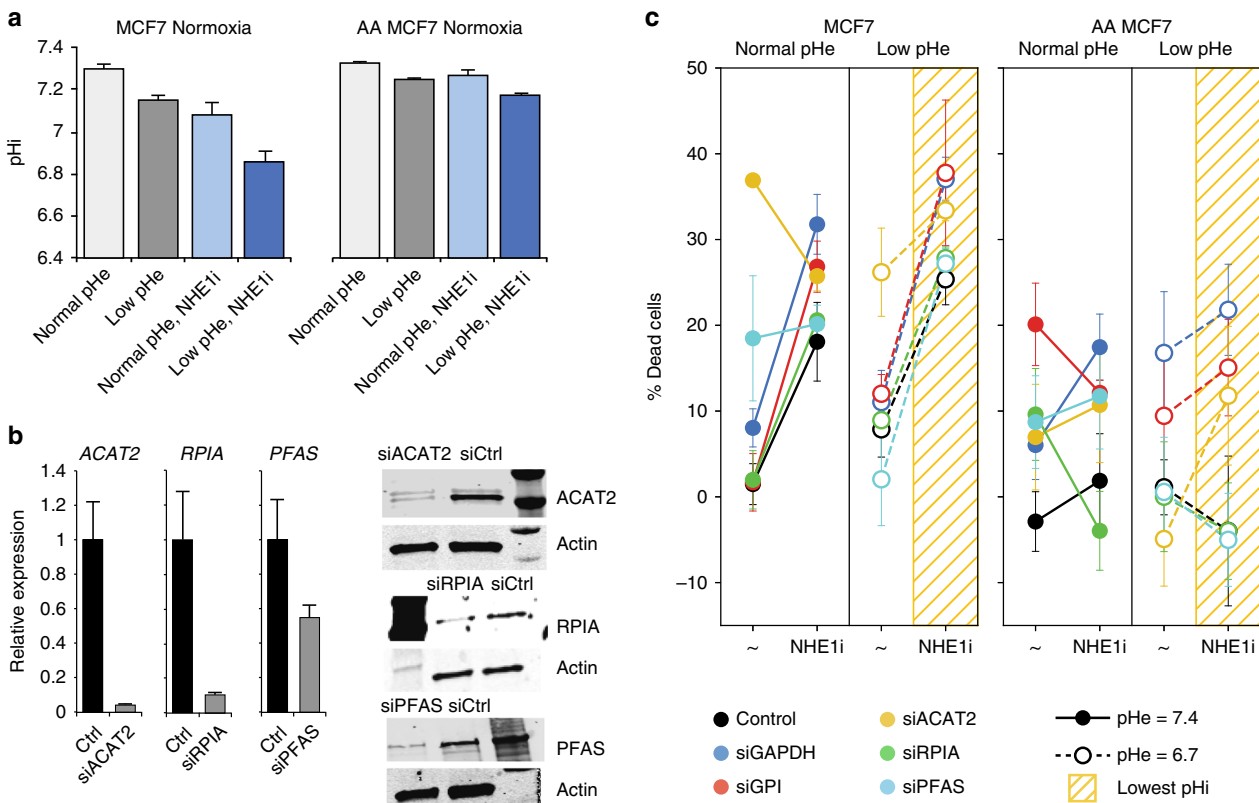

**Fig. 4** Validation of systems analyses in predicting pHi-sensitive metabolic vulnerabilities. **a** pHi measurements of naïve and acid-adapted (AA) breast cancer MCF7 cells, under normoxia and following the inhibition of *NHE1*, by cariporide treatment. For pHi measurments at least 30 cells were analyzed. **b** Efficiency of knockdown of indicated targets at the mRNA and protein levels, following reverse transfection of MCF7 cells with the indicated siRNAs. qPCR was repeated at least three times with three replicates. **c** The effect of gene inhibition in normal and low extracellular pH (pHe) shown for naïve and AA MCF7 breast cancer cells. Similar color code to Fig. 3 is applied. At low pHe where the lowest pHi was obtained there is a large reduction in the viability of cells. In AA cells, only the selective and pH-specific targets (*GAPDH*, *GPI*, and *ACAT2*) achieve amplified anti-proliferative effects following *NHE1* inhibition, despite the smaller reduction in the pHi of these cells. *PFAS*, a selective but not pH-specific target, is similar to control cells following *NHE1* inhibition. Knockdown of *RPIA* had a weak effect in naïve cells and no/opposite effects in AA cells. The viability assay was done three times with four replicates each time. The bars depict the mean and the error bars depict the standard deviation of the mean

## Discussion

Collectively, our findings suggest that cancer cells have superior fitness at an alkaline pHi, and that their reliance on an alkaline intracellular environment confers vulnerabilities that can be exploited for therapeutics. In accord with previous studies[9,26], our findings clearly demonstrate that lowering pHi is a selective vulnerability for cancer cells. Furthermore, here we have established that, with the development of new computational techniques, this vulnerability can be exploited to systematically identify metabolic targets to attack cancer cells at acidic pHi, forming a synthetic lethal therapeutic strategy comprised of targeting transporters that lower the pHi in combination with disabling the selected metabolic targets. Experimental testing of this strategy in breast cancer cell lines indicated that it is particularly effective against cancer cells that have adapted to hypoxia and extracellular acidosis, and that display aggressive phenotypes[21,33,34]. Nonetheless, further study is needed to establish the clinical applicability of the proposed strategy to treat tumor cells, where lowering pHi can require targeting several proton pumps and acid transporters, as in the case of triple negative breast cancer cells[35,36].

Beyond proliferation, our systems analyses also indicated a coupling between intracellular alkalization and the Warburg effect, which is manifest as increased glucose consumption and decreased oxygen uptake rates at high pHi. Accordingly, lowering pHi was predicted by these analyses to reverse to some extent

these adaptations. Further, within the cohort of the identified targets these analyses predicted that disabling *GAPDH* or *GPI* amplifies the anti-Warburg effect of acidic pHi when they are inhibited at acidic pHi, which was then tested experimentally. Interestingly, in parallel to this study, *GAPDH* was recently identified as an anti-Warburg target using other computational and experimental techniques[37], which independently verify the power of integrating the computational analysis and experimental studies reported herein. Nonetheless, *GAPDH* (and *GPI*) is more than a metabolic regulator and has rich functionalities in cancer[38,39], indicating that the exact mechanisms responsible for its potential therapeutic roles remain to be resolved. Our results suggest that the knockdown of *GPI* may have similar if not superior effects, once more potent and selective inhibitors are developed. Other identified targets in our analysis, notably *PGD* and *G6PD* from the pentose phosphate pathway, are also predicted to have both anti-proliferative and anti-Warburg effects on cancer. Hence, in addition to their known anti-cancer roles[40,41], our findings suggest the therapeutic response of tumors to inhibitors of *PGD* and *G6PD* will be amplified at acidic pHi.

Notwithstanding the power of our computational analyses to predict robust network-wide effects of pHi on the metabolic state of cells, some caveats and limitations should be addressed to improve and expand the methodology. First, as biomass production was used as an objective cellular function in optimizing GSMM (Methods), the current analysis reflects only a fraction of

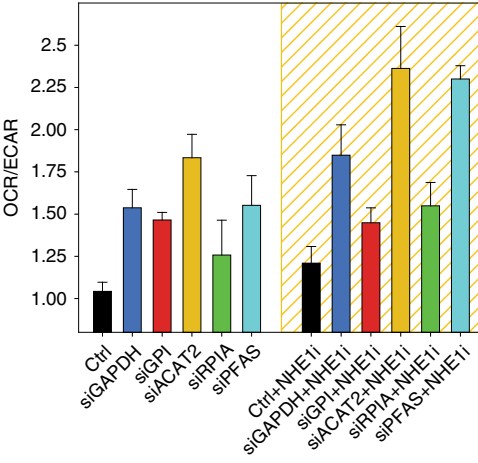

**Fig. 5** Anti-Warburg effects of lowering pHi and inhibition of pHi-dependent metabolic targets. Seahorse flux experiments were preformed for MCF7 naïve cells under normoxia, with (low pHi, yellow grid) and without (physiological pHi) inhibition of *NHE1* and of the indicated metabolic targets. The reduction in pHi (black bars) has an anti-Warburg effect on cancer cell metabolism, as measured by the ratio between the oxygen consumption rate (OCR) and the extracellular acidification rate (ECAR). Both at physiological pHi and at low pHi the additional inhibition of the selected targets (colors), amplifies the anti-Warburg effect on cancer cells. Highest OCR/ECAR ratios are obtained at low pHi (yellow grid). Seahorse experiments were done in six replicas each time and experiments were repeated three times. The bars depict the mean and the error bars depict the standard deviation of the mean

all possible targets, where the formulation of additional objective functions should lead to specific mechanistic insights, and targets that may combat specific phenotypes. Second, to more fully understand the mechanisms that direct metabolic adaptation following perturbations (i.e., lowering pHi and/or gene KO) in a specific cell type, the models should be refined by integration of cell-specific and condition-specific "omics" data. Third, flux-balance analysis (FBA) assumes a metabolic steady state, such that metabolite concentrations are constant in time. However, titrating metabolites is known to affect the pH-dependent behavior of some enzymes[42,43], adding complexity to the prediction and integration of such pH-activity profiles. Thus, improvements of the method should include more explicit considerations of metabolite concentrations, along with the effects of allostery and cooperativity, which require structural biology considerations[44]. Fourth, our knowledge-driven, homology-based pipeline might be improved by distinguishing enzyme isoforms having very different pH-activity profiles[45], and further refined by predicting critical points of half and none activity, where less experimental data are available (Supplementary Figure 1).

Our analysis provides an additional insight into cancer evolution. Ion gradients represent an ancient form of chemiosmotic energy production, observed in hydrothermal vents[46], as well as in a wide range of life forms, from LUCA[47] to bacteria[48]. Such gradients are considered a primitive mechanism relative to respiratory oxidative reactions[46,49,50], and they allow cells to cope with a variety of environmental extremes[51]. Hence, our findings, indicating that intracellular alkalization is coupled to the Warburg effect, may also reflect cancer's general embracing of primitive unicellular strategies for proliferation[2].

Importantly, the computational methodology presented herein extends well beyond the realm of cancer. In particular, pH regulation plays crucial roles in immunity and bacterial antibiotic resistance[52,53], and affects the population diversity and functions

of unicellular communities[54], as well as the function of nerve cells and the brain[55,56]. Hence, this study also provides a computational and conceptual framework for exploring the consequences of pH regulation, and its therapeutic potential across a broad spectrum of biomedical domains.

## Methods

**Reconstruction of pH-activity profiles**. A detailed description of the reconstruction of pH-activity profiles is provided in Supplementary Methods. Briefly, dependency of enzymatic activity on pH was obtained from experimental data in BRENDA (brenda-enzymes.org)[30]. As shown in Fig. 1a, for each enzyme we extracted six critical points, corresponding to the lower (acidic) and upper (alkaline) limits of 0%, 50%, and 100% of activity. Experimental points of 100% activity were mainly obtained from the 'pH Optimum' field in BRENDA. 0% and 50% points were fetched from the 'pH Range' category, after manual curation: records reporting activities up to 25% were approximated to 0%; activities from 25% to 75% were set to 50%; and activities above 75% were set to 100%. When more than one record was available, we extracted the median value.

The vast majority of experimental values corresponded to 100% of activity, i.e., the optimal pH (Supplementary Figure 1). To impute missing critical points, linear regressors were built based on experimental data and values of close homologs. The enzymes in the Recon1 GSMM were then screened against this pH-profile database using JackHMMER[57]. These analyses provided the critical points of enzyme activity for 1444 of the 1905 metabolic genes (76%). To control for over-fitting, our predictions were then validated with three training/test split protocols of increasing merit, i.e., first with a standard 10-fold cross-validation, then by removing all human enzymes from the training sets, and finally by also removing EC information (Supplementary Figures 5–7).

Given the 0%, 50%, and 100% critical pH points, the percentage of activity of an enzyme at any given pH was linearly interpolated. Metabolic enzymes without a predicted profile were conservatively given a constant activity of 100%, such that no constraints were applied to them in the GSMMs.

**Genome-scale metabolic modeling and application of pH-activity profiles**. We applied the pH-activity profiles into a panel of recently developed, data-driven and extensively validated cell-specific GSMMs[27,28], which are based on the human GSMM[58] that is comprised of the NCI-60 cell lines panel (*n* = 60) and the normal healthy lymphoblastic cell lines from the HapMap project (*n* = 224). Briefly, these models integrate gene expression and proliferation data of each cell line to adjust the human metabolic model, by identifying the most significant reactions that correlate with the corresponding phenotypic data of each cell. These panels of models capture key differences between cancer and normal cells, including the Warburg characteristics. Moreover, these models have identical network architecture, number of metabolites and reactions, and are modeled under identical media composition (e.g., DMEM or RPMI-1640); hence, they are ideal for comparative analysis.

Given the need to solve the solution space in each model across a wide range of pHi (6.5–8.5), we selected for analysis from the panels above: (i) a set of eight cancer models, representative of the eight different cancer types that exist in the NCI-60 panel: LE:CCRF-CEM, BR:BT-549, CNS:SF539, CO:HCC2998, RE:ACHN, LC:NCI-H226, OV:OVCAR-5, and ME:SK-MEL-5. Cells were randomly selected from the available subsets of each cancer type in the NCI60 panel; and (ii) a control set of 12 normal cells from the HapMap project panel: 5 Americans with northern and western Europe ancestry, 2 Han Chinese, 2 Japanese, and 3 Yoruba, such that they represent the diversity of subjects in this panel. These sets were sufficient for in silico analysis, as exemplified by the error bars of the simulated rates (Fig. 2, and Supplementary Figures 8–12).

**CBM of metabolic networks**. The CBM approach imposes mass-balance, thermodynamic, and enzymatic capacities constraints to define the allowable functional states of biochemical genome-scale model[59]. These constraints can be mathematically represented as

$$\frac{dx}{dt} = S \cdot v = 0 \qquad (1)$$

$$v_{min} \le v \le v_{max} \qquad (2)$$

where *v* is the network's flux vector and *S* is the *m* × *n* stoichiometric matrix, and where *m* and *n* are the number of metabolites and reactions, respectively. The matrix specifies all biochemical reactions and metabolites in the network. Constraint (1) assures steady state, where the production and consumption rate is equal for each metabolite in the network. Constraint (2) imposes thermodynamic and enzymatic capacities by defining the bounds of the permissible flux of each reaction. In a given metabolic state, the flux that a reaction can carry is then estimated using FBA and flux-variability analysis (FVA)[60], taking its maximal flux as a proxy for its catalytic activity. Similarly, cell proliferation is estimated as the maximal flux carried by the biomass reaction in the GSMM, which represents the cell growth

yield. Since the cells we model are highly proliferative, we constrained cell proliferation (i.e., the objective function) to be larger than 80% of its maximum to infer the activity of all other reactions. The results reported here are robust with respect to the choice of this threshold in the tested range of 70–90% (Supplementary Figure 9).

**Integration of pH-activity profiles into GSMM.** The pH-profiles of Fig. 1 were applied to adjust the bounds of each reaction in the GSMM, at a given pHi, inspired by a similar approach to explore the effects of temperature within the framework of GSMM[61]. This is accomplished in three steps. First, at a given pHi, the activity of genes relative to their maximal activity defines a pH-specific activity of each gene, $W_G = [0, 1]$. Second, considering the $W_G$ of all genes, we infer the activity of all reactions, based on the embedded genes-reactions logical rules that are associated with each reaction. For an "AND" logic the minimal $W_G$ is assumed, and for an "OR" logic the maximal $W_G$ is assumed. Hence, this generates a weight factor, $W_R = [0, 1]$, for each reaction R. Third, the upper and lower bounds of reaction R are scaled by $W_R$. For a bi-directional reactions ($v_{max} \geq 0$ and $v_{min} \leq 0$) the new lower bound is $LB = W_R \times v_{min}$ and the new upper bound is $UB = W_R \times v_{max}$. To avoid invalid ranges of bounds, for a forward reaction ($v_{max} > v_{min} \geq 0$) only the upper bound is scaled by $W_R$, ensuring that $UB \geq v_{min}$. Similarly, for a reverse reaction ($v_{min} < v_{max} \leq 0$) only the lower bound is scaled by $W_R$, ensuring that $LB \leq v_{max}$. We assume that cellular organelles are well buffered and therefore applied these modifications only to cytosolic enzymes. Nonetheless, the results reported here are only weakly insensitive to this choice (Supplementary Figure 12).

**Gene KO simulation and analysis.** The KO of a gene G is simulated by setting $W_G$ to 0–0.1, representing an inhibition of activity of 100–90%, respectively ($W_G = 0$ in the main analysis). The effect of a gene KO on cell proliferation is estimated by $nB_{KO,pH} = B_{KO,pH}/B_{WT,pH}$, where $B_{WT,pH}$ is the biomass of the WT at a given pH, and $B_{KO,pH}$ is the biomass of the cell following gene KO at this pH. To assess the importance of a gene KO, two ranking measures were introduced:

(i) SEL, which measures the difference in cell proliferation between cancer and normal cells following gene KO. Hence, $SEL = nB_{KO,pH}^{Normal} - nB_{KO,pH}^{Cancer}$. The larger SEL the more selective is the gene KO.

(ii) PHS, which measures for a given cell (i.e., normal or cancer cell) how potent the effects of a gene KO are at 'low' pH (pH = 6.7) when compared with its effect at 'physiological' pH (pH = 7.3). Hence, $PHS = nB_{KO,pH=7.3} - nB_{KO,pH=6.7}$. The larger PHS value, the higher is the effect of the gene KO at low pH compared with its effect at the higher pH.

The SEL is evaluated across all cancer–normal pairs ($n = 8 \times 12 = 96$). The PHS is evaluated across all cancer cells ($n = 8$). To avoid numerical precision effects we set any measured flux in each cell and each reaction to zero, if following GSMM optimization the flux rate was below a strict threshold of $|1e-7|$. That is, reported identified targets are those with normalized average scores $>|1e-7|$. Further, a minimal recurrence frequency of 12.5% was set as a minimal threshold, such that at the extreme targets must have non-zero scores in at least one cancer cell type when compared across all normal cells (12/96 for selectivity; 1/8 for pH specificity). The ranking of genes by either SEL or PHS is highly robust within the gene inhibition range tested, $W_G = [0, 0.1]$, and is insensitive to the exact choice of 'low' pH (Supplementary Figure 11).

**Measurement and manipulation of intracellular pH using SNARF-1.** Forty-eight hours before the experiments, 5000 cells were grown onto round-glass bottom 25 mm cell culture dishes in DMEM/F12 medium supplemented with 10% FBS. For hypoxia experiments, culture dishes were transferred to hypoxia machine with 0.1% $O_2$. The day of the experiment media was removed and replaced by fresh DMEM serum-free medium with 5 μM SNARF-1, the pH fluoroprobe. Cellular esterases cleave the succinimidyl ester groups leaving the charged free-acid form of SNARF-1 in the cytosol. For loading of SNARF-1, cells were incubated for 30 min at 37 °C followed by three washes of DPBS. Fluorescence images of cells were obtained using 40× and 63× objective using oil immersion lenses. SNARF-1 was excited at 534 nm, and emission signal was collected at 580 nm (long bandpass filter) and 640 nm. For short-term hypoxia (1–30 min) pHi was measured before hypoxia and every 5 min after hypoxia using a confocal microscope equipped with a hypoxia chamber and $CO_2$ supply at 37 °C. After background subtraction, the 640 and 580 nm channels are used to measure ratiometric pHi, as specified by the manufacturer.

For in situ calibration, SNARF-1 loaded cancer cells were incubated with 10 μM nigericin in the presence of 100 mM $K^+$ to equilibrate the intracellular pH with the controlled extracellular medium. Calibration in living cells removes light dispersion side effects. Images were then collected as extracellular pH is varied with the same instrument settings for all calibrations and experiments. To evaluate the effect of long-term hypoxia and MCT1/2 inhibitor treatment on steady-state cytosolic pH, cells growing on round-glass bottom culture plates were treated with or without inhibitor under normoxic and hypoxic conditions (0.1% $O_2$) for 48–72 h. To measure the effects of acidosis on pHi, cells were incubated in low pH (6.7) media for 72 h and pHi was measured using SNARF-1. Cells grown under physiological pH (7.4) were used as a control. Intracellular pH was calculated by the formula pH

$= pK_a - \log[(R-R_{max})/(R_{min}-R)]$; wherein $R$ is the measured 580/640 fluorescence ratios, and $pK_a$, $R_{min}$, and $R_{max}$ were determined to be respectively, 7.30, 2.54, and 0.56, from the in situ calibration curves for MCF7 breast cancer cells.

To evaluate the effect of MCT1/2 and NHE1 inhibitor treatment on steady-state cytosolic pHi of naïve and AA MCF7 cells, and of naïve and AA MDA-MB-231 breast cancer cells, the cells were grown in glass bottom culture plates (25 mm) and incubated for 24 h. Media was then replaced with fresh media containing 1 μM MCT1/2 or 10 μM NHE1 inhibitor, and plates were placed in 37 °C incubator under hypoxia (0.1% $O_2$) and normoxia as control. The data were from three independent experiments, each performed in triplicate and with at least 30 single cells/plate. Data are shown as mean and the error of the mean (standard deviation).

**siRNA transfection.** Breast cancer cells were seeded at 5000 cells per well in 96-well plates or 500,000 cells in a six-well plate 24 h before transfection. Two different sets of siRNA from two different companies (validated siRNA from Thermofisher and Dharmacon) were used to knockdown the targets. GAPDH, GPI, PFAS, ACAT2, RPIA, or negative control siRNAs were transfected using Lipofectamine RNAiMAX (Invitrogen) and a reverse transfection technique. In brief, reverse transfection siRNA/Lipofectamine complexes in serum-free media were loaded into the wells and cells were then added to them to promote efficiency of transfection. 4 h after transfection, media with 10% FBS was added. Following transfection cells were incubated in 37 °C in normoxia (20% $O_2$) or hypoxia (0.1% $O_2$) and normal pHe (7.4) or low pHe (6.5) while treated with MCT1/2 or NHE1 inhibitors to reduce the pHi.

**Western blot analysis.** To validate the efficiency of siRNA knockdown at the protein level, and to assess the status of MCT1 and MCT4 expression in naïve and acid adapted MCF7 breast cancer cells western blots were performed. Cells transfected with siRNAs were harvested 48–72 h after transfection and lysed in RIPA buffer containing 1× protease inhibitor cocktail (Sigma-Aldrich). 20 μg of protein per sample was loaded on polyacrylamide–SDS gels that were then blotted onto nitrocellulose. Membranes were incubated with primary antibodies against GAPDH (Cat# 2118 Cell Signaling, 1:2000), GPI (ab68643, Abcam, 1:1000), ACAT2 (Cat# 13294s Cell Signaling 1:1000), RPIA (ab181235, abcam, 1:500), PFAS (PA554628, Thermofisher, 1:200), MCT1 (sc-365501, Santa Cruz Biotechnology, 1:500), MCT4 (sc-376140, Santa Cruz Biotechnology, 1:500), and β-Actin (A5441, Sigma, 1:6000). Odyssey fluorescence system and chemiluminescence were used for membrane development. Proteins detected ran at the expected molecular weights, as verified using molecular weight standard markers. Uncropped western blots that were used for the data presented in Fig. 3b, Fig. 4b, Supplementary Figure 15B, and Supplementary Figure 16B are provided in Supplementary Figure 18. Western blot analyses were repeated at least twice.

**qRT-PCR analysis.** To confirm the efficiency and selectivity of siRNA-mediated knockdown, cells were harvested 48 h post-transfection and RNA was extracted using a RNA extraction kit (Qiagen). GAPDH-specific primer sets were as follows: forward, 5′-CTGGCATCATGTATTTAGGGGC-3′; and reverse, 5′-GAGTTGCGGCCTGTCA GAAAC-3′. GPI primer sets were as follows: forward, 5′- TCGCCCAACCAACTC TATTG-3′; reverse, 5′-GATGATGCCCTGAACGAAGAT-3′. β-Actin was used for normalization of PCR results. β-Actin primer sets were as follows; forward, 5′-CG GCATCGTCACCAACTG-3′; reverse, 5′-GGCACACGCAGCTCATTG-3′. RPIA-specific primer sets were as follows: forward, 5′-AGTGCTGGGAATTGGAAGT GG-3′; reverse, 5′- GGGAATACAGACGAGGTTCAGA-3′. PFAS-specific primer sets were as follows; forward, 5′-CCCAGTCCTTCACTTCTATGTTC-3′; reverse, 5′-GTAGCACAGTTCAGTCTCGAC-3′. ACAT2-specific primer sets were as follows: forward, 5′-GCGGACCATCATAGGTTCCTT-3′; reverse, 5′-ACTGGCTTGTCTCA ACAGGATTCT-3′. The qRT-PCR experiments were repeated twice with at least three replicas each time.

**Proliferation studies.** Cells treated with siRNA were seeded at $1 \times 10^5$/ml in six-well plates in triplicate and counted on an Invitrogen cell counter following trypan blue dye staining to determine the number of living and dead (blue) cells. Briefly, cells cultured for the indicated intervals under the four growth conditions (normoxia, pH 7.4; normoxia, pH 6.7; hypoxia (0.1% Oxygen), pH 7.4; hypoxia (0.1% oxygen), pH 6.7) were trypsinized and diluted in their growth media. A filtered 0.4% trypan blue dye solution was prepared and added 1:1 to count the cells with an Invitrogen Cell Counter. siRNA treatment was renewed after each round of cell counting (every 72 h). Proliferation experiments were repeated three times with at least two replicates for each sample. The identity of the cancer cell lines used in these studies was confirmed by STR analyses performed by the Molecular Genomics Core of the Moffitt Cancer Center.

**Viability assays.** Cell viability was measured after 72 h post treatment with target siRNAs using Cell Counting Kit-8 (CCK-8) under the four growth conditions and following transfection with siRNAs for GPI, GAPDH, PFAS, RPIA, ACAT2, or siCtrl and with and without treatment with the MCT1/2 inhibitor SR-13800[35]. CCK8 is a sensitive colorimetric-based viability assay based on Dojindo's highly water-soluble tetrazolium salt, with WST-8 as its active agent. CCK8 was used to measure viability as it is not pH sensitive and can be added to the cells directly in

their niche, without fixation or change of media. For measuring viability, cells were seeded in a 96-well plate (with triplicate of the same samples), and viability was measured at the indicated intervals. The experiments were repeated three times.

**Glycolytic and OCR measurements**. Glycolytic rate of MCF7 and AA-MCF7 cancer cells treated with siRNAs and NHE1 inhibitor was measured using Seahorse XF96 extracellular flux analyzer and a glycol-rate kit (Seahorse Biosciences). OCR and ECAR of cancer cells were determined by seeding them on XF96 microplates in their growth medium until they reached over 90% confluence. In these studies, seeding started with 10,000 cells (50% of well area) and reverse transfection was applied. Measurements were determined 48–72 h later when the cells reached the 90% confluence. 1 h before the seahorse measurements culture media were removed and cells were washed three times with PBS following by addition of base medium (non-buffered DMEM supplemented with 25 mM glucose). For glycolytic rate measurements, mitochondria inhibitors including rotenone (1 μM) and anti-mycin A (1 μM), were injected after basal measurements of ECAR and OCR of the cells under treatment to stop the mitochondrial acidification. 2-deoxy-glucose (100 mM) was added next to bring down glycolysis to basal levels. Finally, data were normalized for total protein content of each well using the Bradford protein assay (Thermofisher). Seahorse measurements were performed with 4–6 technical replicates and these experiments were repeated four times.

**Code availability**. All the analysis was done in MATLAB 2016b under academic license to UMD/UMIACS/CBCB. MATLAB files, including the algorithm which integrates pHi profiles into the GSMMs, the cell line models used in this study, as well as analysis scripts which reproduce the in silico results are provided as Supplementary Software.

**Data availability**. pH profiles were obtained from BRENDA. Human GSMMs of cancer and normal cell lines were obtained from ref. [28] (https://elifesciences.org/articles/03641/figures#SD4-data). Human metabolic enzymes in the human genome scale metabolic model, and their inferred pHi profiles are provided in Supplementary Data 1. Any additional data is available upon request from the authors.

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

## Acknowledgements

We thank Adam Weinstock and Keren Yizhak for assistance with genome-scale metabolic modeling, and the staff of the Molecular Genomics Core and Analytical Microscopy Core of the Moffitt Cancer Center for their assistance with these studies. We also thank Chunying Yang and Weimin Li for technical assistance, and Nir Gonen, Matthew Oberhardt, Tamir Epstein, Robert Gatenby, Eyal Gottlieb, Kira Makarova, Yuri I. Wolf, Michael Galperin, and Eugene V. Koonin for insightful discussions. Finally, we thank Allon Wagner, Alon Silberman, Edoardo Gaude, Christoph Kaleta, Daniel J. Dwyer, Ayelet Erez, and Christian Frezza for critical reading of early versions of the manuscript and valuable feedback. The study was supported by the University of Maryland Institute for Advanced Computer Studies (UMIACS), the Israeli I-CORE Program, grants from the Israeli Science Foundation (ISF), the EU FP7 INFECT project, NIST and IPST to E.R. P.A. was supported by grants from ERC (614944 SysPharmAD), EU (306240 SyStem-Age), and MINECO (BIO2013-44222-R). J.L.C. and W.R.R. were supported by a grant from NCI (CA154739). J.L.C. also received support from NCI Cancer Center Support Grant P30 CA076292 and from the Cortner-Couch Chair for Cancer Research from the University of South Florida School of Medicine. R.J.G. was supported by a grant from NCI (R01CA077571). E.P. was partially supported by the Terrapin project of UMD and M.D.-F. was partially supported by scholarships from EMBO, FEBS, and Spanish FPU.

## Author contributions

E.P., M.D.-F. and E.R. conceived the study. E.P. and M.D.-F. developed the methodology and performed the computational analysis. M.D. designed and performed the experimental work. P.A. and E.R. supervised the computational work. R.J.G. and J.L.C. supervised the experimental work. W.R.R. synthesized the MCT1/2 inhibitor. All authors participated in the design of the study and interpretation of the results. E.P., M.D.-F., J.L.C. and E.R. wrote the paper with contributions from all authors. All authors have read and approved the final version of the manuscript. E.P. and E.R. jointly directed the study.

## Additional information

**Competing interests:** The authors declare no competing interests.

