## [Peer Review File · Nature Communications]

Reviewers' comments:

Reviewer #1 (Remarks to the Author):

Persi et al., present a new approach by using computational analysis based on pH optima of enzyme activities to understand how the higher intracellular pH of cancer cells might affect the activity of metabolic enzymes and consequently metabolic reprogramming. The idea is exciting and like all good studies would open new directions for investigation. However, conclusions of the work are overstated because limited experimental data verify computational predictions. An example of a new, potentially significant idea based on their computational analysis is that metabolic enzymes in some pathways (pentose phosphate, glycolysis, fatty acid oxidation) but not others (amino acid metabolism) have activities regulated within the cellular pH range. This prediction should be validated for a few key enzymes in each pathway or by pathway readouts in cells, but validation is not included beyond measuring glucose uptake and lactate production. In addition to limited experimental validation, another concern is how study is presented/promoted without acknowledging caveats and modulation by the native environment compared with in vitro measurements of pH optima of enzyme activities.

Presentation and conclusions

1. The authors' interpretation and conclusions are sometimes overstated or misleading.

- Computational analysis can suggest but not determine how pHi modulates metabolism, as indicated in the abstract. The computational profiles presented are predictive. Although experimental data are included to test some predictions, collectively the work suggests but does not reveal how pHi modulates metabolism.
- The statements on p5 of no difference in pH-dependence of NADPH production between cancer and normal cells and that metabolism is changing "independently of redox" are overreaching and misleading. Other reducing equivalents (GSH/GSSG) could be changing. Moreover, the sharp decrease in NADPH at pHi>7.2 in their model suggests that cancer cells have a distinctly different redox environment compared with normal cells, even if availability of all other reducing equivalents are ignored.
- The first sentence of the Summary - these findings establish that cancer cells ... should be revised.
- The claim of synthetic lethality requires better controls with MCF10A studies, showing statistically significant GAPDH or GP1 knockdown, and confirming when pHi does not change - which is the crux of the argument for synergy between MCT inhibition and k.d. of GAPDH/GP1 or hypoxia.

2. The premise that pH-dependent metabolism is predominantly determined by pH optima and enzyme homologies is presented rather simplistically without discussing caveats of allostery, cooperativity, modulation and effects of titrating metabolites.

- Critical pH optimum determined in vitro does not necessarily indicate enzyme activity in the native environment. One example is phosphofructokinase-1 activity, which may have a recognized pH optima but pH regulated activity is dependent on ATP concentrations - i.e. a higher pH can overcome inhibition of high ATP. This is particularly relevant to the authors' statement on p6 on the high ATP production rate in cancer.
- Homologous enzymes having similar critical pH values (Fig. S2-S4 and a component of the profile and search) should include some circumspection because computational predictions of function based on homology can be inaccurate (ACS Chem Biol 2016 11(9):2420). Again, using PFK1 as an example, the ability of increased pH to relieve inhibition by ATP is isoform specific and mostly relevant to muscle PFK1 and not at all to the platelet isoform, despite these isoforms having a high degree of sequence homology.
- Overall, pH-dependent effects on metabolism are much more complex than merely pH optima of metabolic enzyme activities, and the manuscript should acknowledge and discuss this. Recent findings indicate pH effects on metabolism can also be regulated by titration of metabolites (JBC

2016 291(38):20188 and Nat Chem Biol 2017 13(5):494).

3. A significant amount of description/discussion of database/training set/etc prep is in supplemental material. This may be due to length constraints of the main text, but incorporating some into the main manuscript, particularly aspects of the SI background, would improve understanding.

Data and analysis

1. Data on accurately imaging pHi with hypoxia are new and significant – which require a microscope with a hypoxia chamber. These data are a distinct strength.

2. Computational data in Fig. 2 on predicted selectivity of pH sensitive metabolic pathways are very interesting but would have stronger impact with supporting experimental validation. Only glucose uptake and extracellular lactate are determined, which is not sufficient to indicate glycolytic flux or support the statement in the Summary “our finding that intracellular alkalization is coupled to the Warburg effect.” Relative to the breast cancer lines used for Fig. 3, do pHi manipulations affect predicted pH sensitive enzymes or readouts for the PP and FAO pathways but not amino acid metabolism?

3. Inhibiting the lactate transporter to lower pHi applies a metabolic stress, not merely a change in pHi, but is referred to as merely a pH stress. Hypoxia or knocking down expression of metabolic enzymes likely enhances metabolic stress on an already metabolically stressed cell. An additional approach such as inhibiting ion transporters or pumps regulating pHi or using available optogenetic reagents (caged H⁺ donors or archaeorhodopsins) is important.

4. Stated on p8. “In acid adapted cancer cells, hypoxia +MCT1/2 inhibition achieved a reduction of pH of about 0.3 units. These results were independent of pHe levels (Figure S13).” However, Figure S13 does not show pHi measured in hypoxic cells at normal and acidic pHe as it does for normoxic cells.

5. Statistics and number of replicates are not sufficiently indicated. These are particularly important for evaluating GAPDH and GP1 knockdown and consequences. For Fig 2 it is unclear, even after reading the methods, what statistical cut-offs the authors are using to determine ‘pH-specific’ and ‘selective’. On p25 (odds ratio of a right-tailed Fisher’s exact test of 995.3, pvalue ~0), the Fisher’s exact gives p values that should be included.

Additional comments

In the Abstract, consequences of increased intracellular pH being poorly understood is not entirely accurate. Perhaps incompletely understood but not “poorly” because there are numerous publications on increased proliferation, migration, survival and also a few on tumorigenesis.

In the Background, ref 8 and 9 are cited to support “reverse pHi gradient correlating with the spatial gradient of oxygen availability.” However, ref. 8 shows that these two parameters are not correlated spatially (vs mean pHi correlating with mean O₂ availability), and ref. 9 does not measure pHi but shows only that O₂ gradient correlating with expression of ion transporters.

Fig. 2. p5, in silico analyses can possibly predict lower oxygen consumption and increased glucose uptake, but not “reveal” as stated in the text.

p6 The authors state that the low effect of the k.o.s on ROS generation is indicative of the low probability of the cells developing hypermutated and resistant phenotypes, which is not sound logic. Other components (even components of the metabolism (2HG) can induce resistant and hypermutated phenotypes in cells. Moreover, hypermutation can be induced by mutated repair proteins that don’t require any change in ROS.

p30. "Training and test sets were split from the very initial data (i.e. before homology-based imputation". This is unclear. The reason for splitting training and test sets at that point vs. other point should be clarified.

p38. SNARF does not allow measurements of pKa. Perhaps this is a typo and should state pH_i.

Reviewer #2 (Remarks to the Author):

In the manuscript 'Systems Analysis of Intracellular pH Vulnerabilities for Cancer Therapy', Persi et al. report a computational model of the pH-dependence of metabolism using flux balance analysis. Intracellular pH was quantitatively modeled by integrating pH-specific enzymatic activities in the BRENDA database with constraint-based genome-scale metabolic models. These pH-dependent models were used in predicting anti-proliferation targets effective under lower pH values. Generally speaking, the idea of combining machine learning, enzyme parameters and GSMMs in identifying pH-dependent vulnerabilities in cancer cells appears novel and important to me. The experimental validation of the computational predictions could be clearer and the rationale for focusing on lower pH_i needs to be explained more clearly. Several major points are listed below.

Major:

1. The presentation of experimental results could be clearer. For instance, the theoretical predictions are some dependence of proliferation fold change on pH_i with knockdown of the predicted targets, but this is not easy to see from the growth rates reported in Fig 3. A better visualization of the results could be some scatter plots showing how growth rate fold change varies with measured intracellular pH.

2. In fig 1C, enzymes whose optimal pH is available in the BRENDA database and those whose values were estimated by missing data imputation should be shown separately. It is not surprising that for enzymes whose optimal pH values are already known, the optimal pH values are close to the physiological pH of the cellular compartments they locate in. It would be a stronger prediction if the authors could show that predicted optimal pH values (not those measured values available in BRENDA) also correlate with the pH of corresponding compartments.

3. Some details about the GSMMs are necessary for the readers to fully understand the computational modeling. According to their earlier publications, there are some individualized GSMMs for NCI-60 cell lines but I'm not sure which one is used in this paper. How the models for normal cells are made is also unclear to me. If the authors could provide more details on these models, that will be helpful.

Minor:

Pg 15, the last paragraph: 'MCF7 calls' should be 'MCF7 cells'.

Reviewer #3 (Remarks to the Author):

The authors have developed the systems to analyse pH roles in cancer metabolism and provided a conceptual and computational framework for exploring pH roles across biology.

1. I think It is an overstatement, the current work can not be treated as a foundation for across ALL biology

The authors have generated pH activity profiles for metabolic enzymes by extracting the complete

record of experimental measurements of the activity of all enzymes at different pH across all taxa using BRENDA

2. It is not clear the relevance of all taxa to human metabolism especially to understand cancer? The authors claimed that they have systematically identified metabolic targets whose inhibition selectively kills cancer cells at acidic pH, forming a novel therapeutic strategy.

3. Most of genes, identified by this method as potential anticancer therapy, have been already reported in literature.

3.1 GAPDH/GAPDHS is not a novel target. Glyceraldehyde-3-phosphate dehydrogenase (GAPDH) is a glycolytic enzyme specifically catalyzing the reversible conversion of glyceraldehyde-3-phosphate (G-3-P) to 1,3-diphosphoglycerate. GAPDH participates in numerous cellular functions, in addition to glycolytic effects. For instance, GAPDH contributes to nuclear tRNA export, DNA replication and repair, endocytosis, exocytosis, cytoskeletal organization, iron metabolism, carcinogenesis, and cell death [1]

3.2 It was known that 6PGD is involved in cancer cell metabolism so 6PGD inhibitors have been sought [2].

3.3 GPI was reported as an important gene in glioblastoma [3]

3.4 G6PD was known as a target and biomarker for cancer [4]

4. Statistical analysis of predictions was adequate, and the work is reproducible.

Overall, it's not clear how this method could predict completely novel therapeutically targets. It would be good if authors will work more on completely new targets or combinations of existing ones.

References

[1] Critical protein GAPDH and its regulatory mechanisms in cancer cells. Zhang JY, Zhang F, Hong CQ, Giuliano AE, Cui XJ, Zhou GJ, Zhang GJ, Cui YK. *Cancer Biol Med*. 2015 Mar;12(1):10-22. doi: 10.7497/j.issn.2095-3941.2014.0019. Review.PMID: 25859407

[2] 6-Phosphogluconate dehydrogenase links oxidative PPP, lipogenesis and tumour growth by inhibiting LKB1-AMPK signalling. 2015

[3] Silencing of phosphoglucose isomerase/autocrine motility factor decreases U87 human glioblastoma cell migration. Li Y, et al. *Int J Mol Med*, 2016 Apr. PMID 26936801, Free PMC Article

[4] Glucose-6-phosphate dehydrogenase: a biomarker and potential therapeutic target for cancer. Zhang C, Zhang Z, Zhu Y, Qin S1. *Anticancer Agents Med Chem*. 2014 Feb;14(2):280-9.

Reply to Reviewers

Reviewer #1

Persi et al., present a new approach by using computational analysis based on pH optima of enzyme activities to understand how the higher intracellular pH of cancer cells might affect the activity of metabolic enzymes and consequently metabolic reprogramming. The idea is exciting and like all good studies would open new directions for investigation. However, conclusions of the work are overstated because limited experimental data verify computational predictions. An example of a new, potentially significant idea based on their computational analysis is that metabolic enzymes in some pathways (pentose phosphate, glycolysis, fatty acid oxidation) but not others (amino acid metabolism) have activities regulated within the cellular pH range. This prediction should be validated for a few key enzymes in each pathway or by pathway readouts in cells, but validation is not included beyond measuring glucose uptake and lactate production. In addition to limited experimental validation, another concern is how study is presented/promoted without acknowledging caveats and modulation by the native environment compared with in vitro measurements of pH optima of enzyme activities.

We thank the Reviewer for the laudatory comments as well as for his/her detailed and constructive review. All the concerns noted are insightful and important, and we believe that addressing these significantly improved our manuscript. In the revised manuscript we have addressed all of these concerns, including additional validation studies that support predictions from our systems analyses, as explained below point-by-point.

Presentation and conclusions

1. The authors' interpretation and conclusions are sometimes overstated or misleading.

- Computational analysis can suggest but not determine how pHi modulates metabolism, as indicated in the abstract. The computational profiles presented are predictive. Although experimental data are included to test some predictions, collectively the work suggests but does not reveal how pHi modulates metabolism.

As correctly noted by the Reviewer, computational analyses are only predictive, and we believe this is well emphasized throughout the text, as well as by separating the systems/computational and experimental Subsections of the **Results** section. We have changed the Abstract accordingly, and in addressing other comments of the Reviewers we stress even further this point. The Abstract now reads:

“A reverse pH-gradient is a hallmark of cancer metabolism, manifested by extracellular acidosis and intracellular alkalization. While consequences of extracellular acidosis are known, the roles of intracellular alkalization are incompletely understood. By reconstructing and integrating enzymatic pH-dependent activity profiles into cell-specific genome-scale metabolic models, we developed a computational methodology that explores how intracellular pH (pHi) modulates metabolism. We show that in-silico, alkaline pHi maximizes cancer proliferation coupled to increased glycolysis and adaptation to hypoxia (i.e., Warburg effect), whereas acidic pHi disables these adaptations and compromises tumor cell growth. We then systematically identified metabolic targets (GAPDH and GPI) with predicted

amplified anti-cancer effects at acidic pHi, forming a novel therapeutic strategy. Experimental testing of the strategy in breast cancer cells revealed that it is particularly effective against aggressive phenotypes. Hence, this study suggests essential roles of pHi in cancer metabolism and provides a conceptual and computational framework for exploring pHi roles in other biomedical domains.”

- The statements on p5 of no difference in pH-dependence of NADPH production between cancer and normal cells and that metabolism is changing “independently of redox” are overreaching and misleading. Other reducing equivalents (GSH/GSSG) could be changing. Moreover, the sharp decrease in NADPH at pHi>7.2 in their model suggests that cancer cells have a distinctly different redox environment compared with normal cells, even if availability of all other reducing equivalents are ignored.

We thank the Reviewer for raising this issue. We agree and have modified the text accordingly, which now reads “...presumably with mild effect on redox”. To further clarify, within the framework of GSMM, metabolite concentrations are constant. Therefore, we cannot estimate the redox state by measuring ratios such as GSH/GSSH, and therefore we use the overall NADPH production rate as an indicator. Furthermore, the mild effect of changing pHi on ROS production is likely indicative of mild effects on redox state. This was emphasized in **Methods** and is now also further discussed in the revised **Discussion**, where we now provide a subsection entitled **Caveats and Limitations** of our systems/computational analyses (please see below).

- The first sentence of the Summary - these findings establish that cancer cells ... should be revised.

We, agree. We have substantially revised the text here and elsewhere in the **Discussion**. Specifically, in this sentence we substituted “*establish*” with “*suggest*”.

- The claim of synthetic lethality requires better controls with MCF10A studies, showing statistically significant GAPDH or GPI knockdown, and confirming when pHi does not change – which is the crux of the argument for synergy between MCT inhibition and k.d. of GAPDH/GPI or hypoxia.

We measured the effect of MCT1/2 inhibitors – and now an NHE1 inhibitor – with and without GAPDH or GPI knockdown, across all conditions, including in MCF10A breast epithelial cells [see **Figure S14**]. We describe these results in the main text in the section **Experimental Proof of concept**, where we have previously written:

*“Finally, consistent with the model, the strategy is selective to cancer cells, as there were only very modest effects of hypoxia, MCT1/2 inhibition and knockdown of GAPDH or GPI on the survival of normal MCF10A breast epithelial cells (**Figure S14**)”.*

2. The premise that pH-dependent metabolism is predominantly determined by pH optima and enzyme homologies is presented rather simplistically without discussing caveats of allostery, cooperativity, modulation and effects of titrating metabolites.

- Critical pH optimum determined *in vitro* does not necessarily indicate enzyme activity in the native environment. One example is phosphofructokinase-1 activity, which may have a recognized pH optima

but pH regulated activity is dependent on ATP concentrations – i.e. a higher pH can overcome inhibition of high ATP. This is particularly relevant to the authors’ statement on p6 on the high ATP production rate in cancer.

- Homologous enzymes having similar critical pH values (Fig. S2-S4 and a component of the profile and search) should include some circumspection because computational predictions of function based on homology can be inaccurate (ACS Chem Biol 2016 11(9):2420). Again, using PFK1 as an example, the ability of increased pH to relieve inhibition by ATP is isoform specific and mostly relevant to muscle PFK1 and not at all to the platelet isoform, despite these isoforms having a high degree of sequence homology.
- Overall, pH-dependent effects on metabolism are much more complex than merely pH optima of metabolic enzyme activities, and the manuscript should acknowledge and discuss this. Recent findings indicate pH effects on metabolism can also be regulated by titration of metabolites (JBC 2016 291(38):20188 and Nat Chem Biol 2017 13(5):494).

We agree that our computational pipeline has caveats and limitations. Allosterity, cooperativity and titrating metabolites, isoforms with different sensitivities to pH and fundamental assumptions of constraint-based modeling, are all important limitations of the current methodology, and are relevant directions for future follow up studies. As suggested by the Reviewer, we have substantially expanded our **Discussion**, and included a dedicated subsection titled **Caveats and Limitations** in the revised manuscript. This subsection now reads:

“Notwithstanding the power of our computational analyses to predict robust network-wide effects of pH_i on the metabolic state of cells, some caveats and limitations should be addressed to improve and expand the methodology. First, as biomass production was used as an objective cellular function in optimizing GSMM (Methods), the current analysis reflects only a fraction of all possible targets, where the formulation of additional objective functions should lead to specific mechanistic insights and targets that may combat specific phenotypes. Second, to more fully understand the mechanisms that direct metabolic adaptation following perturbations (i.e., lowering pH_i and/or gene knockout) in a specific cell type, the models should be refined by integration of cell-specific and condition-specific “omics” data. Third, flux-balance analysis assumes a metabolic steady state, such that metabolite concentrations are constant in time. However, titrating metabolites is known to affect the pH-dependent behavior of some enzymes (42-43), adding complexity to the prediction and integration of such pH-activity profiles. Thus, improvements of the method should include more explicit considerations of metabolite concentrations, along with the effects of allosterity and cooperativity, which require structural biology considerations (44). Fourth, our knowledge-driven, homology-based pipeline might be improved by distinguishing enzyme isoforms having very different pH-activity profiles (45), and further refined by predicting critical points of half and none activity, where less experimental data are available (Figure S1).”

Furthermore, we would like to clarify several important points regarding our computational pipeline and analyses, emphasizing that, even when considering its limitations; it is robust with respect to following points:

- Because pH optimum is not sufficient to model enzymatic dependence on pH, we invested great efforts in curating and predicting the critical pH points of half (50%) and loss of (0%) activity. Therefore, our analysis considers not only the optima of activity, but rather the full activity curve.
- It is true that *in-vitro* measurements of enzymatic activities in BRENDA might not fully correspond to physiological conditions. Nonetheless, we reassuringly observed a remarkable correlation of our predicted optimal pH and the pHi of the corresponding cellular compartments (**Figure 1C**), suggesting that, in general, there is a correspondence between BRENDA *in-vitro* measurements and physiological conditions.
- Overall, our results are robust to significant variability in the pH-activity curves and vanish under wrong assignments of profiles to enzymes (**Figure S9**). This robustness analysis indicates that the pH-dependent behavior analyzed in our study is a global property that is best explained by the network as a whole, involving the cumulative behavior of many enzymes, and that it is less sensitive to the exact details of the pH-activity curve of a given enzyme.
- Supporting **Figures S2-S4** demonstrate that homologous proteins tend to have similar pH-activity profiles, and our computational pipeline capitalizes on this observation. Furthermore, specifically for the human metabolic network enzymes (Recon-1), 80.5% of the enzymes were available in BRENDA and, therefore, sequence data was not necessary for these enzymes. For the other 19% of the enzymes, we found records with the same EC number in other species, and sequence homologies were only then used to weight the averaging of profiles. Indeed, we didn't find the corresponding EC number for only 0.5% of the metabolic enzymes, and these were found through a purely sequence-based search. These statistics are now added to **Supplementary Methods**.

3. A significant amount of description/discussion of database/training set/etc prep is in supplemental material. This may due to length constraints of the main text, but incorporating some into the main manuscript, particularly aspects of the SI background, would improve understanding.

We thank the Reviewer for this suggestion. Accordingly, we have now extended the subsection **Computational pipeline** of the **Results**, as well as the **Methods**, to address this point. In the **Results** section we added the following statements:

(i) "Intracellular pH fluctuations affect enzyme activity by modifying protonation states of key catalytic residues and compromising stability of structural folds (29). Thus, to model the effects of pHi on the metabolic state of cells, it is essential to know the pH-dependent activity profile of each enzyme. Fortunately, elucidating enzymatic pH-activity profiles is a classical task of enzymologists, who need this to optimize the experimental conditions of their assays. This knowledge has been accumulated in the scientific literature over the years and databases like BRENDA (30) are devoted to cataloguing it."

(ii) "This knowledge-based approach was superior to more classical physics-based methods that are focused on predicting pH-stability (Figure S3)".

In **Methods** section, we have added the following statement:

"To control for over-fitting, our predictions were validated with three train/test split protocols of increasing merit, i.e. first with an standard 10-fold cross-validation, then by removing all human enzymes from the training sets, and finally by also removing EC information (Figures S5-S7)."

Data and analysis

1. Data on accurately imaging pHi with hypoxia are new and significant – which require a microscope with a hypoxia chamber. These data are a distinct strength.

We thank the Reviewer for this positive comment.

2. Computational data in Fig. 2 on predicted selectivity of pH sensitive metabolic pathways are very interesting but would have stronger impact with supporting experimental validation. Only glucose uptake and extracellular lactate are determined, which is not sufficient to indicate glycolytic flux or support the statement in the Summary “our finding that intracellular alkalization is coupled to the Warburg effect.” Relative to the breast cancer lines used for Fig. 3, do pHi manipulations affect predicted pH sensitive enzymes or readouts for the PP and FAO pathways but not amino acid metabolism?

We thank the Reviewer for suggesting these additional experimental studies, which we believe substantially strengthen proof-of-concept of our systems analyses.

Firstly, in the revised manuscript we have modified the computational analysis, and distinguish between the anti-proliferative (**Figure 2A**) and anti-Warburg (**Figure 2B**) effects of intracellular acidosis, and the enhancement of these effects by inhibiting the targets identified by our computational analysis at low pHi. Further, **Figure 2C** now shows the amplified predicted **anti-proliferative** effects of inhibiting *selective* and *pH-specific* targets, and **Figure 2D** now shows the amplified **anti-Warburg** effects of the knockdowns of these targets (as measured by the ratio OCR to ECAR) at low pHi. We have modified the text in subsection ***In-Silico* Analysis of pH-dependent metabolism** to account for these revisions.

Secondly, we now provide two new experiments that were designed to further validate the anti-proliferative effects (new **Figure 4**) and the anti-Warburg effects (new **Figure 5**) of the proposed strategy (i.e., intracellular acidosis + targeted therapy), which address the following major concerns raised by the Reviewer. The following results are now described in the revised subsection **Experimental Proof of Concept**:

- In the first set of experiments we measured the viability of MCF7 breast cancer cells, with and without knockdown of selected targets, using an independent different technique to manipulate the pHi, by inhibiting the NHE1 channel under normoxia (as opposed to chronic hypoxia + MCT1/2 inhibition), which addresses the Reviewer’s concern regarding other stresses imposed by hypoxia (i.e., Reviewer point 3 below). In these experiments we tested naïve and acid-adapted MCF7 cells, in two external conditions, low and physiological pHe. As before, we silenced GAPDH and GPI, and also tested 3 additional targets, RPIA from the pentose-phosphate pathway, ACAT2 from the fatty-acid metabolism pathway, and PFAS from the amino-acid biosynthesis pathway. The results of these experiments are now shown in a new Figure 4. They are concordant with our previous studies targeting GAPDH and GPI + MCT1/2 inhibition (Figure 3). Specifically, these studies show that the largest anti-proliferative effects at low pHi, here driven by inhibition of NHE1, are obtained for the *pH-specific* targets GAPDH, GPI and to a lesser extent ACAT2, in agreement with its lower scores, while the inhibition of the *non-pH-specific* target PFAS was less effective, with comparable effects to the control at low pHi (i.e., lack of amplified anti-proliferative effect). The selective and pH-specific

target *RPIA* showed inconsistent results, where it elicited some effect in naïve cells but not in acid-adapted cells. This may be due to an insufficient low pHi to fully inhibit this enzyme. Finally, these results further indicate that targeting GAPDH and GPI would be specifically effective against aggressive (e.g., acid adapted) cancer phenotypes.

These results are now described in the main text, and read as follows:

“As hypoxia may impose stress on cells and elicit network-wide effects that are hard to control and measure, additional experiments were performed to assess the robustness of the results in naïve and acid-adapted MCF7 cells (Figure 4). In these experiments pHi under normoxia was reduced by inhibiting the Na⁺-H⁺ exchanger NHE1 via treatment with cariporide. This strategy was most successful at low pHe (Figure 4A). We then inhibited the top targets (GAPDH and GPI), as well as three additional targets identified from different metabolic pathways (RPIA, ACAT2 and PFAS). The efficiency of siRNA-directed knockdown of these targets was verified by qRT-PCR and western blot analyses (Figure 4B). Also in these experiments, knockdown of these targets compromised the survival of MCF7 cancer cells at low pHi, and again this was more significant at low pHe, where the lowest pHi was achieved (Figure 4C). Across the different conditions and cells, the predicted selective and pH-specific targets GAPDH, GPI, and ACAT2 achieved the largest detrimental effect on cancer cell survival. Relative to control, this was more pronounced in acid-adapted cells, despite the smaller reduction in pHi in these cells, suggesting these targets play important roles in the aggressive phenotypes manifest in these cells. In contrast, the predicted selective and pH-specific target RPIA displayed inconsistent effects across cells, possibly because it requires larger reduction in pHi to induce anti-proliferative effects in acid-adapted cells. Finally, as an important control, knockdown of the predicted selective but not pH-specific target PFAS did not amplify the anti-proliferative effect relative to control cells at low pHi, and is the least pHi-sensitive metabolic target.”

- In the second set of experiments, we performed Seahorse measurements to assess the anti-Warburg effect of lowering pHi and of inhibiting selected targets, by quantifying changes in the ratio of OCR to ECAR. These results are now added as a new Figure 5. These studies revealed that lowering pHi by treatment with the NHE1 inhibitor drives the metabolic state of cancer cells to a more oxidative and less fermentative state, supporting our systems analyses presented in **Figure 2B**. Further, the additional inhibition of all the 5 targets examined amplified this anti-Warburg shift in cell metabolism, where this is more significant at low pHi, supporting our analyses presented in **Figure 2D**. Nonetheless, we have found that the model underestimated the effect of intracellular acidosis, as relative to controls the inhibition of all examined targets has an amplified anti-Warburg effect on cancer cells (in both physiological pHi and low pHi), while the model predicted this only for a few targets.

These results are now described in the main text, and read as follows:

“Finally, to validate the anti-Warburg effect of lowering pHi and of inhibiting selected pHi-dependent targets, we performed Seahorse experiments and measured the anti-Warburg ratio (OCR/ECAR) in MCF7 breast cancer cells (Figure 5). Lowering pHi alone was revealed to have an anti-Warburg effect on cancer cells, consistent with our computational results (cf. Figure 2B). Further, the knockdown of each of the pHi-dependent targets amplified the anti-Warburg effect on

cancer cells at physiological pHi, and this amplification was more significant at low pHi, following NHE1 inhibition, also consistent with our model (cf. Figure 2D). Nonetheless, while the model predicted an amplified effect relative to control cells for only a few targets, the experiments show that all of the targets examined exhibited an amplified anti-Warburg effect, across all conditions. Thus, intracellular acidosis has a stronger anti-Warburg effect than that predicted by the model. This discrepancy may be due to a number of differences between the model and experiments, as discussed below.”

Finally, as the Reviewer raised an interesting point regarding the regulation/role of pathways, we would like to clarify that the association of a target with a pathway does not necessarily imply the regulation of the entire pathway. A dedicated computational-experimental study under such complex manipulations is required to elucidate the regulation and role of different pathways, where one would need to also integrate expression, proteomics and activity-based proteomic profiling data (measured across all conditions and genetic manipulations) into a genome-scale metabolic model of a specific cell. Both the computational analysis as well as corresponding validating experiments for such analyses require each different techniques. This is thus beyond the scope of the current study, but we agree is very interesting and we plan to embark on it in a future study.

3. Inhibiting the lactate transporter to lower pHi applies a metabolic stress, not merely a change in pHi, but is referred to as merely a pH stress. Hypoxia or knocking down expression of metabolic enzymes likely enhances metabolic stress on an already metabolically stressed cell. An additional approach such as inhibiting ion transporters or pumps regulating pHi or using available optogenetic reagents (caged H⁺ donors or archaeorhodopsins) is important.

We have addressed these concerns in two ways. **First**, with regard to our previous results (**Figure 3**), we now better describe that we used chronic hypoxia, which presumably imposes less stress on the cells as opposed to acute hypoxia. **Second**, and more importantly, we now introduced an additional technique to manipulate pHi under normoxia by inhibiting the NHE1 channel. These new experimental results (**Figure 4**) are concordant with those obtained using chronic hypoxia plus MCT1/2 inhibition (**Figure 3**). Please see our detailed reply to point 2 above.

4. Stated on p8. “In acid adapted cancer cells, hypoxia +MCT1/2 inhibition achieved a reduction of pH of about 0.3 units. These results were independent of pHe levels (Figure S13).” However, Figure S13 does not show pHi measured in hypoxic cells at normal and acidic pHe as it does for normoxic cells.

We have added these measurements to Figure S13.

5. Statistics and number of replicates are not sufficiently indicated. These are particularly important for evaluating GAPDH and GP1 knockdown and consequences. For Fig 2 it is unclear, even after reading the methods, what statistical cut-offs the authors are using to determine ‘pH-specific’ and ‘selective’. On p25 (odds ratio of a right-tailed Fisher’s exact test of 995.3, pvalue ~0), the Fisher’s exact gives p values that should be included.

- Replicas Stat: Each experiment has at least two biological replicates, each including 3 to 4 technical replica. All the stats are now added to each relevant subsection of the **Methods**.
- Computed selectivity and pH-specificity scores: We thank the Reviewer for these suggestions, and we agree this indeed required a better description. We now report that all targets that have non-zero score (beyond a conservative numerical error of $|10e-7|$), and require that the scores recur with a minimal frequency of 12.5% among all the cells examined. Given the design of the cohort of models (8 cancer types x 12 reference normal cells = 96 pair comparisons) this requirement corresponds to demanding that, at the extreme, at least one cancer cell type shows a consistent behavior when compared to all 12 normal cells. These considerations are now described in **Methods** (subsection **Gene knockout simulation and analysis**). The frequency of the scores is now shown in the revised **Figure 2C** for each target.
- P-values: We put 0 *P*-value when the exact *P*-value is below the numerical precision of our computers. To avoid confusion, we now report a *P*-value $< 1e-323$ in such cases.

Additional comments

In the Abstract, consequences of increased intracellular pH being poorly understood is not entirely accurate. Perhaps incompletely understood but not “poorly” because there are numerous publications on increased proliferation, migration, survival and also a few on tumorigenesis.

We agree, and changed the word “poorly” to “*incompletely understood*” as suggested.

In the Background, ref 8 and 9 are cited to support “reverse pHi gradient correlating with the spatial gradient of oxygen availability.” However, ref. 8 shows that these two parameters are not correlated spatially (vs mean pHi correlating with mean O2 availability), and ref. 9 does not measure pHi but shows only that O2 gradient correlating with expression of ion transporters.

We have modified this section to more accurately describe these results, and we refer and cite a recent review on this matter (ref **10**, Corbet & Feron, *Nat Rev Cancer*, 2017). This revised text now reads:

“Although locally highly diverse, the mean pHe and oxygen pressure (pO2) both decrease in a highly correlated manner with the distance from nearest blood vessels in tumors (8). The latter evokes changes in the activity of various transporters promoting intracellular alkalization (9), with an overall significant correlation, yet a non-linear relationship, between the reverse pH gradient and oxygen availability (10)”.

Fig. 2. p5, in silico analyses can possibly predict lower oxygen consumption and increased glucose uptake, but not “reveal” as stated in the text.

We agree. This is now addressed with the additional Seahorse experiments that we provide, which support our systems analyses presented in **Figure 2B** and **2D**. Please see our reply to point 2 above (in “Data Analysis”).

p6 The authors state that the low effect of the k.o.s on ROS generation is indicative of the low probability of the cells developing hypermutated and resistant phenotypes, which is not sound logic. Other components (even components of the metabolism (2HG) can induce resistant and hypermutated phenotypes in cells. Moreover, hypermutation can be induced by mutated repair proteins that don't require any change in ROS.

This is true of course. However, we did not claim that any possible mechanism leading to hypermutation or resistance to treatment in cancer is accounted for by ROS. Rather, we stated that the risks associated with high ROS levels are not amplified by these knockdowns. We have modified this sentence to avoid confusion: It now reads:

“Moreover, the overall predicted mild effect of these knockouts on the production rate of reactive oxygen species (ROS) indicates that they are not likely to induce risks associated with excess ROS levels in cancer, namely ROS-induced hypermutation and resistant phenotypes (30)”

p30. “Training and test sets were split from the very initial data (i.e. before homology-based imputation”. This is unclear. The reason for splitting training and test sets at that point vs. other point should be clarified.

- We thank the Reviewer for this suggestion. The revised sentence (Supplementary Methods, subsection **Validation**) now reads: *“To ensure complete independence between training and test sets in the cross-validation, and to avoid over-fitting the data, training and test sets were split from the raw experimental data that we compiled from BRENDA, i.e. before the homology-based imputation step.”*
- We have also added the following sentence in **Methods**: *“To control for over-fitting, our predictions were validated with three training/test split protocols of increasing merit, i.e. first with an standard 10-fold cross-validation, then by removing all human enzymes from the training sets, and finally by also removing EC information (Figures S5-S7).”*

p38. SNARF does not allow measurements of pKa. Perhaps this is a typo and should state pH_i.

We didn't measure pKa with SNARF. We correct the pKa of SNARF inside the cells for more accurate pH_i measurements. SNARF1 pKa outside the cells and in its solvent is 7.5. However because of light dispersion in biological environment the pKa is different inside of the cells and even varies with different cell types. Therefore we correct the pKa in the target cells each time we do new experiment. Finally the pH_i is measured from the spectra of SNARF with correlated pKa inside the cells using the formula below. This is now added to **Methods** section, subsection **Measurement and manipulation of intracellular pH using SNARF-1**.

$$\text{pH} = \text{pK}_A - \log \left[\frac{R - R_B}{R_A - R} \times \frac{F_B(\lambda_2)}{F_A(\lambda_2)} \right]$$

Reviewer #2

In the manuscript ‘Systems Analysis of Intracellular pH Vulnerabilities for Cancer Therapy’, Persi et al. report a computational model of the pH-dependence of metabolism using flux balance analysis. Intracellular pH was quantitatively modeled by integrating pH-specific enzymatic activities in the BRENDA database with constraint-based genome-scale metabolic models. These pH-dependent models were used in predicting anti-proliferation targets effective under lower pH values. Generally speaking, the idea of combining machine learning, enzyme parameters and GSMMs in identifying pH-dependent vulnerabilities in cancer cells appears novel and important to me. The experimental validation of the computational predictions could be clearer and the rationale for focusing on lower pH needs to be explained more clearly. Several major points are listed below.

We thank the Reviewer for the positive feedback. In the revised manuscript we have addressed all of the concerns raised, as noted below point-by-point.

Major

1. The presentation of experimental results could be clearer. For instance, the theoretical predictions are some dependence of proliferation fold change on pHi with knockdown of the predicted targets, but this is not easy to see from the growth rates reported in Fig 3. A better visualization of the results could be some scatter plots showing how growth rate fold change varies with measured intracellular pH.

We thank the Reviewer for this suggestion. Please note that it is not feasible to measure pHi in each and every experiment/manipulation, hence we cannot plot the results as function of pHi. Nonetheless, we have devoted considerable effort to present these results in a more comprehensive and clearer manner. In the revised manuscript we have changed the format of Figure 3 (and new Figure 4) and present the effect of pHi using connected lines (and used symbols to address the effect of different pHe conditions, and colors to relate to the different genetic perturbations). In the new plots, the effect of lowering pHi is clearly depicted by the changes in cell proliferation and survival following MCT1/2 or NHE1 inhibition, and conditions where the lowest pHi was achieved are highlighted with a yellow background grid.

2. In fig 1C, enzymes whose optimal pH is available in the BRENDA database and those whose values were estimated by missing data imputation should be shown separately. It is not surprising that for enzymes whose optimal pH values are already known, the optimal pH values are close to the physiological pH of the cellular compartments they locate in. It would be a stronger prediction if the authors could show that predicted optimal pH values (not those measured values available in BRENDA) also correlate with the pH of corresponding compartments.

We thank the Reviewer for this important suggestion. We have performed this test and the data are now shown in revised Figure 1C. We still observe a good correspondence between cellular compartment pH and *predicted* optima of activity. In the figure, the “Including experiments” boxes correspond to the pH optima that were used in the subsequent GSMM modeling. As a validation, we include “predictions only” boxes, which are the result of the 10-fold cross-validation (presented in full in Figure S5).

3. Some details about the GSMMs are necessary for the readers to fully understand the computational modeling. According to their earlier publications, there are some individualized GSMMs for NCI-60 cell

lines but I'm not sure which one is used in this paper. How the models for normal cells are made is also unclear to me. If the authors could provide more details on these models, that will be helpful.

We substantially expanded the description of the cell-specific models and the selected cohorts for analysis, at the beginning of the subsection **Genome-scale metabolic modeling and application of pH-activity profiles** in the **Methods** section. Further, we note that these *in-silico* models (both of cancer and normal cells) have been constructed using the same method, which has been extensively described in our previous publications (Yizhak et al. *eLife*, 2014, and Yizhak et al. *Mol Sys Biol*, 2014) which we cite. Hence, readers can find both further details as well as the reconstructed models in these references. The modified text in the **Methods** section now reads:

“We applied the pH-activity profiles into a panel of recently developed, data-driven and extensively validated cell-specific genome-scale metabolic models (GSMMs) (27-28), which are based on the human GSMM (58) that is comprised of the NCI-60 cell lines panel (n=60) and the normal healthy lymphoblastic cell lines from the HapMap project (n=224). Briefly, these models integrate gene expression and proliferation data of each cell line to adjust the human metabolic model, by identifying the most significant reactions that correlate with the corresponding phenotypic data of each cell. These panels of models capture key differences between cancer and normal cells, including the Warburg characteristics. Moreover, these models have identical network architecture, number of metabolites and reactions, and are modeled under identical media composition (e.g., DMEM or RPMI-1640); hence, they are ideal for comparative analysis.

Given the need to solve the solution space in each model across a wide range of pHi (6.5 to 8.5), we selected for analysis from the panels above: (i) a set of 8 cancer models, representative of the 8 different cancer types that exist in the NCI-60 panel: LE:CCRF-CEM, BR:BT-549, CNS:SF539, CO:HCC2998, RE:ACHN, LC:NCI-H226, OV:OVCAR-5, and ME:SK-MEL-5. Cells were randomly selected from the available subsets of each cancer type in the NCI60 panel; and (ii) a control set of 12 normal cells from the HapMap project panel: 5 Americans with northern and western Europe ancestry, 2 Han Chinese, 2 Japanese and 3 Yoruba, such that they represent the diversity of subjects in this panel. These sets were sufficient for in-silico analysis, as exemplified by the error bars of the simulated rates (Figure 2, and Figures S8-S12).”

Minor

Pg 15, the last paragraph: ‘MCF7 calls’ should be ‘MCF7 cells’.

Thanks and corrected.

Reviewer #3

The authors have developed the systems to analyze pH roles in cancer metabolism and provided a conceptual and computational framework for exploring pH roles across biology.

1. I think it is an overstatement; the current work cannot be treated as a foundation for across ALL biology.

We of course agree, and did not mean to imply that. We do believe however that the approach presented may be implemented to study the role of pHi in other biological systems of interest, as GSMMs are available for a number of unicellular species (including fungi and bacteria), the methods exist to construct cell-specific GSMMs and the pH-profiles can be queried from our constructed database (see new **Perspectives** subsection in the **revised Discussion**). We have, however, toned down the pertinent text, and in the **revised Abstract** this text now reads: “...provides a conceptual and computational framework for exploring pHi roles in other biomedical domains”.

The authors have generated pH activity profiles for metabolic enzymes by extracting the complete record of experimental measurements of the activity of all enzymes at different pH across all taxa using BRENDA

2. It is not clear the relevance of all taxa to human metabolism especially to understand cancer?

- Other taxa were used to allow prediction and imputations of missing data, based on the high similarity of pH critical point across similar EC categories, and were considered only when human data were missing. In practice, however, most of the profiles are based on existing human data in BRENDA: Specifically, for the human metabolic network enzymes, 80.5% of the enzymes were available in BRENDA and, we only used data for the other 19% of the enzymes (with the same EC number) in other species, and used sequence-homolog statistics to fine-tune our predictions based on BRENDA. Only 0.5% of the enzymes lacked a corresponding EC number and here we found matches through a purely sequence-based search. This is now added and clearly described in **Supplementary Methods**.
- Further, we stress that we have shown through various measures that both the predictions of the pH-profiles (**Figures S1-S7**) and well as the *in-silico* metabolic behaviors (**Figures S8-S12**) are highly robust. In particular we have shown that: **(i)** our predictions of critical pH values perform well even without consideration human information at all (**Figure S6**); and **(ii)** the average network-wide behavior *in-silico* is not sensitive to the exact details of the pH-profiles (**Figure S8**). Hence, we believe that information from other taxa can be indeed relevant and useful to infer critical pH values (when not available), and therefore we used these in our pipeline, though it was not critical to our analyses.
- Last, all caveats of the computational pipeline are now stated in a dedicated paragraph of the Discussion, as there were similar concerns expressed by Reviewer #1. Please see above (**reply to Reviewer #1**) the new subsection of **Caveats and limitation** of the **revised Discussion**.

The authors claimed that they have systematically identified metabolic targets whose inhibition selectively kills cancer cells at acidic pHi, forming a novel therapeutic strategy.

3. Most of genes, identified by this method as potential anticancer therapy, have been already reported in literature.

3.1 GAPDH/GAPDHS is not a novel target. Glyceraldehyde-3-phosphate dehydrogenase (GAPDH) is a glycolytic enzyme specifically catalyzing the reversible conversion of glyceraldehyde-3-phosphate (G-3-P) to 1,3-diphosphoglycerate. GAPDH participates in numerous cellular functions, in addition to glycolytic effects. For instance, GAPDH contributes to nuclear tRNA export, DNA replication and repair, endocytosis, exocytosis, cytoskeletal organization, iron metabolism, carcinogenesis, and cell death [1].

3.2 It was known that 6PGD is involved in cancer cell metabolism so 6PGD inhibitors have been sought [2].

3.3 GPI was reported as an important gene in glioblastoma [3]

3.4 G6PD was known as a target and biomarker for cancer [4]

We thank the Reviewer for noting these facts. We are well aware that these targets themselves are not novel. Importantly, we didn't claim that they are new targets, but rather that their inhibition and anti-proliferative and anti-Warburg effects on cancer cells can be markedly amplified at low pHi, forming a new strategy based on lowering the pHi to increase the efficacy and selectivity of agents that disable these targets. Further we note that finding that a given target is important in a specific cancer type (as noted by the Reviewer in point 3.3) does not necessarily mean that it would work in another cancer type, specifically in the context of having an amplified deleterious effect on cancer at low pHi and/or anti-Warburg effects. In the revised manuscript we have substantially expanded our **Discussion**, clarifying these points, and cite most of the studies noted by the Reviewer, including a very recent study on the anti-Warburg effect of GAPDH (Liberti et al, *Cell Metabolism*, 2017). The beginning of the **revised Discussion** (followed by discussion on *caveats and limitations*) now reads:

“Collectively, our findings suggest that cancer cells have superior fitness at an alkaline pHi and that their reliance on an alkaline intracellular environment confers vulnerabilities that can be exploited for therapeutics. In accord with previous studies (9, 26), our findings clearly demonstrate that lowering pHi is a selective vulnerability for cancer cells. Furthermore, here we have established that, with the development of new computational techniques, this vulnerability can be exploited to systematically identify metabolic targets to attack cancer cells at acidic pHi, forming a synthetic lethal therapeutic strategy comprised of targeting transporters that lower the pHi in combination with disabling the selected metabolic targets. Experimental testing of this strategy in breast cancer cell lines indicated that it is particularly effective against cancer cells that have adapted to hypoxia and extracellular acidosis and that display aggressive phenotypes (21, 33-34). Nonetheless, further study is needed to establish the clinical applicability of the proposed strategy to treat tumor cells, where lowering pHi requires targeting several proton pumps and acid transporters, as in the case of triple negative breast cancer cells (35, 36).

Beyond proliferation, our systems analyses also indicated a coupling between intracellular alkalization and the Warburg effect, which is manifest as increased glucose consumption and decreased oxygen uptake rates at high pHi. Accordingly, lowering pHi was predicted by these analyses to reverse to some extent these adaptations. Further, within the cohort of the identified targets these analyses predicted that disabling GAPDH or GPI amplifies the anti-Warburg effect of acidic pHi when they are inhibited at acidic pHi, which was then tested experimentally. Interestingly, in parallel to this study, GAPDH was recently identified as an anti-Warburg target using other computational and experimental techniques (37), which independently verify the power of integrating the computational analysis and experimental studies reported herein. Nonetheless, GAPDH (and GPI) is more than a metabolic regulator and has rich functionalities in cancer (38-39), indicating that the exact mechanisms responsible for its potential

therapeutic roles remain to be resolved. Our results suggest that the knockdown of GPI may have similar effects, if not superior effects, once more potent and selective inhibitors are developed. Other identified targets in our analysis, notably PGD and G6PD from the pentose phosphate pathway, are also predicted to have both anti-proliferative and anti-Warburg effects on cancer. Hence, in addition to their known anti-cancer roles (40-41), our findings suggest the therapeutic response of tumors to inhibitors of PGD and G6PD will be amplified at acidic pHi.”

4. Statistical analysis of predictions was adequate, and the work is reproducible. Overall, it's not clear how this method could predict completely novel therapeutically targets. It would be good if authors will work more on completely new targets or combinations of existing ones.

The focus of the current study was to develop a new strategy for a pHi-dependent combination attack on cancer cells. Applying this approach, it turns out that emerging key predictions indeed involve a few recently reported metabolic targets, but their association and increased effectiveness in low pHi has not been identified before, and we believe this is novel and important on its own. Importantly however, responding to related concerns of Reviewer #1 we have now included a new set of experiments in the revised manuscript (**Figure 4** and **5**), where we inhibited additional metabolic enzymes in combination with the inhibition of the acid transporter NHE1. The response to Reviewer #1 reads as follows:

“... we also tested 3 additional targets, RPIA from the pentose-phosphate pathway, ACAT2 from the fatty-acid metabolism pathway, and PFAS from the amino-acid biosynthesis pathway. The results of these experiments are now added as a new **Figure 4**. They are concordant with our previous results (**Figure 3**). They show that the largest anti-proliferative effects at low pHi are obtained for the *pH-specific* targets *GPADH*, *GPI* (and to a lesser extent *ACAT2*, in agreement with its lower scores), while the inhibition of the not *pH-specific* target *PFAS* was less effective, having an effect comparable to the control at low pHi (i.e., lack of amplified anti-proliferative effect) and is less pHi-sensitive than *pH-specific* targets. The selective and pHi-specific target *RPIA* showed inconsistent results, where it elicited some effect in naïve cells but not in acid-adapted cells. This may be due to the insufficient low pHi obtained in these cells. They results also further indicate that *GAPDH* and *GPI* are specifically effective against aggressive phenotypes.”

References

- [1] Critical protein GAPDH and its regulatory mechanisms in cancer cells. Zhang JY, Zhang F, Hong CQ, Giuliano AE, Cui XJ, Zhou GJ, Zhang GJ, Cui YK. Cancer Biol Med. 2015 Mar;12(1):10-22. doi: 10.7497/j.issn.2095-3941.2014.0019. Review.PMID: 25859407
- [2] 6-Phosphogluconate dehydrogenase links oxidative PPP, lipogenesis and tumour growth by inhibiting LKB1-AMPK signalling. 2015
- [3] Silencing of phosphoglucose isomerase/autocrine motility factor decreases U87 human glioblastoma cell migration. Li Y, et al. Int J Mol Med, 2016 Apr. PMID 26936801, Free PMC Article
- [4] Glucose-6-phosphate dehydrogenase: a biomarker and potential therapeutic target for cancer. Zhang C, Zhang Z, Zhu Y, Qin S1. Anticancer Agents Med Chem. 2014 Feb;14(2):280-9.

Reviewers' comments:

Reviewer #1 (Remarks to the Author):

The revision by Persi et al., is improved by being more circumspect and including some new data. New data using cariporide to lower pHi (with their previous approach of inhibiting MCTs) strengthens the significance of the work and more strongly suggests that the metabolic, proliferation, and death effects are due to pHi and not metabolic stress. However, the revision is not improved for readability. As the previous review indicated, the manuscript is dense, confusing to read, and lacks sufficient clarity to effectively convey the concept to a broad readership. It also retains a mixed message on proliferation vs metabolism – these 2 processes cannot be separated as simplistically as the authors present.

The Abstract states “explores how intracellular pH (pHi) modulates metabolism” but should be revised to “explores how intracellular pH (pHi) can modulate metabolism”

Synthetic lethality of inhibiting H⁺ efflux with oncogene expression was previously shown but is not cited (eLife 4:e03270. PMID:25793441). Additionally, including Seahorse data with MCF10A cells would more strongly support the claim of synthetic lethal effect of lowering pHi and the anti-Warburg effect observed in MCF7 cells. Is this is unique to cancer cells or are all cells susceptible to lower pHi/higher oxphox?

Reviewer #2 (Remarks to the Author):

The authors have done a satisfactory job in addressing my comments.

Reviewer #3 (Remarks to the Author):

comments are addressed and paper could be published. More experiments needed to validate the approach on pathway and tissue level

Reviewer #1

The revision by Persi et al., is improved by being more circumspect and including some new data. New data using cariporide to lower pHi (with their previous approach of inhibiting MCTs) strengthens the significance of the work and more strongly suggests that the metabolic, proliferation, and death effects are due to pHi and not metabolic stress.

We thank the reviewer for these comments.

However, the revision is not improved for readability. As the previous review indicated, the manuscript is dense, confusing to read, and lacks sufficient clarity to effectively convey the concept to a broad readership. It also retains a mixed message on proliferation vs metabolism – these 2 processes cannot be separated as simplistically as the authors present.

In this revision we addressed every comment of the reviewer regarding the readability of the paper. Specifically, the computational part of the paper is now organized in two subsections: (1) the data construction (i.e. pH profiles), and (2) the computational simulations (i.e. genome-scale metabolic modeling), which well separates between the effects of pHi on proliferation (Figure 2A) and on the metabolic state of cells (Figure 2B). Similarly, the effects of KO on proliferation (Figure 2C) are now separated from their effects on the metabolic state (Figure 2D). Also the experimental proof of concept distinguishes between the effects on proliferation (and viability) (Figure 3-4; with two techniques) and the effects on the metabolic state (Seahorse experiments) (Figure 5). To further clarify this, we have modified in the section “In-silico analysis” the pertaining text, which now reads:

“These analyses also predict that the effect of pHi on proliferation is coupled to the metabolic state of cells, whereby lower oxygen consumption and increased glucose uptake rates are observed in cancer cells at high pHi, while at low pHi these adaptations are reversed (Figure 2B). As oxygen is available to all cells ex vivo, this suggests a fundamental coupling between the Warburg effect and intracellular alkalization in cancer cells, consistent with the understanding that the Warburg effect supports proliferation (2).”

The Abstract states “explores how intracellular pH (pHi) modulates metabolism” but should be revised to “explores how intracellular pH (pHi) can modulate metabolism”

We have changed this accordingly.

Synthetic lethality of inhibiting H⁺ efflux with oncogene expression was previously shown but is not cited (eLife 4:e03270. PMID:25793441).

This reference is a key reference in our paper (ref. 26), which is referred both in the introduction and discussion.

Additionally, including Seahorse data with MCF10A cells would more strongly support the claim of synthetic lethal effect of lowering pHi and the anti-Warburg effect observed in MCF7 cells. Is this is unique to cancer cells or are all cells susceptible to lower pHi/higher oxphox?

We have done additional experiments to respond to this comment. In the revised manuscript we have added Seahorse experiments with MCF10A cells as well as pHi measurement and viability assays. These

results are now included as new **Figure S17**. They show that these cells are not sensitive to the inhibition of NHE1 and that the OCR/ECAR doesn't change either (unlike MCF7 cells).

Reviewer #2

The authors have done a satisfactory job in addressing my comments.

We thank the reviewer for this comment and his previous report.

Reviewer #3

Comments are addressed and paper could be published. More experiments needed to validate the approach on pathway and tissue level.

We thank the reviewer for this comment and his previous report.